# QA-Calibration of Language Model Confidence Scores

**Putra Manggala** *
University of Amsterdam
putra.manggala@gmail.com

**Atalanti Mastakouri, Elke Kirschbaum & Shiva Prasad Kasiviswanathan**
Amazon

**Aaditya Ramdas**
Amazon and Carnegie Mellon University

## ABSTRACT

To use generative question-and-answering (QA) systems for decision-making and in any critical application, these systems need to provide well-calibrated confidence scores that reflect the correctness of their answers. Existing calibration methods aim to ensure that the confidence score is *on average* indicative of the likelihood that the answer is correct. We argue, however, that this standard (average-case) notion of calibration is difficult to interpret for decision-making in generative QA. To address this, we generalize the standard notion of average calibration and introduce QA-calibration, which ensures calibration holds across different question-and-answer groups. We then propose discretized posthoc calibration schemes for achieving QA-calibration. We establish distribution-free guarantees on the performance of this method and validate our method on confidence scores returned by elicitation prompts across multiple QA benchmarks and large language models (LLMs).

## 1 INTRODUCTION

Language models (LMs) built on transformer-based architectures are capable of producing texts that are both coherent and contextually relevant for a large range of applications (Brown et al., 2020; Chowdhery et al., 2023; Achiam et al., 2023). In question-and-answering (QA) systems, these models generally perform well, but occasionally produce inaccurate answers – a phenomenon generally referred to as *hallucination* (Huang et al., 2023). Confidence estimates that are paired with the answers can be used as an interpretable indicator of the LM's accuracy (Steyvers et al., 2024). But for this, these confidence scores have to be well-calibrated, i.e., match the actual accuracy of the model.

To evaluate whether the obtained confidence scores are actually well-calibrated, a common criterion is the expected (average-case) calibration error (Tian et al., 2023; Xiong et al., 2024). Suppose a model claims that its answer has a confidence of $p$. Based on only this one answer, it is not possible to know whether this confidence was well-calibrated or not. But when considering multiple question-answer pairs–say $N_p$ pairs each with a claimed confidence $p$, we can verify how many answers are actually correct and measure the error between the claimed confidence and the model's accuracy. Averaging over these errors quantifies of the model's confidence scores.[1]

While this average-case calibration measure makes sense for models trained and evaluated on specific tasks, its applicability to generative QA is questionable because it averages over all QA pairs. This is because generative QA systems can be applied across various domains and topics, such as geography, politics, or medicine. Consider, for instance, the QA pairs shown in Figure 1. On average, this model has a calibration error of $0.5$. From User 1's perspective, the calibration is much worse, with an error of $0.8$. Meanwhile, User 2 has a completely different experience, finding the model to be

---

*Work partially done when author was an intern at Amazon.
[1]Note that the accuracy of the LM is itself unaffected by calibration, as calibration does not change the weights of the LM model.

| User 1 (interested in countries) | User 2 (interested in politics) |
|---|---|
| Question: What timezone is Toronto, Canada?
Answer: CET
Confidence: 0.8
Label: Incorrect | Question: Does Singapore have a prime minister?
Answer: Yes
Confidence: 0.8
Label: Correct |
| Question: Which currency is used in Hungary?
Answer: Lira
Confidence: 0.8
Label: Incorrect | Question: How many years has Justin Trudeau been a prime minister?
Answer: 8
Confidence 2: 0.8
Label: Correct |
| Calibration error for User 1 = 0.8 | Calibration error for User 2 = 0.2 |

Figure 1: Two users interact separately with an LM by inputting questions and obtaining answers and confidence scores from the LM. The LM could be calibrated on average across user types, but each individual user may not have calibrated confidence scores. See Example 2.1 for more details.

much better calibrated than indicated by the average calibration error. This motivates our notion of *QA-calibration*, where the calibration target is conditional on the *group of the question-and-answer pair*.

Previous works have explored group-wise calibration for classifiers, using pre-specified groupings of their covariates (e.g., race or gender) (Kleinberg et al., 2017; Pleiss et al., 2017). However, it is not clear how this idea can be transplanted to the generative QA setting.

A common approach to obtain calibrated confidence scores in LMs is to use confidence elicitation via prompting (Tian et al., 2023; Xiong et al., 2024). The advantage of this approach is that it can be executed with only black-box access to LMs.[2] However, the issue with a pure elicitation via prompting approach is that the performance is sensitive to choice of prompts and model (Sclar et al., 2024), and it does not have any rigorous calibration guarantees. These issues can be mitigated by performing a posthoc calibration of the elicited confidence scores. Our proposed approach is such a posthoc calibration method.

A limitation of relying on LM-elicited confidence directly, or posthoc calibrated using temperature scaling (Tian et al., 2023), is that the output probability is not discretized, making performance difficult to assess (Kumar et al., 2019). We overcome this problem by developing posthoc QA-calibration schemes on elicited confidence scores that use ideas of (histogram) *binning* (Zadrozny & Elkan, 2001) and *scaling-binning* (Kumar et al., 2019) that are by construction discretized.

**Our Contributions:** We make the following contributions in this paper:

1. We introduce QA-calibration, a principled and interpretable notion of calibration in generative QA. Crucially, it defines a fixed mapping $\beta$ from all possible question–and–answer pairs to a finite set. Our QA-calibration notion *generalizes* the standard average-case calibration notion by requiring calibration *conditional on $\beta$*. Due to this conditioning on $\beta$, the guarantee of QA-calibration is stronger than the standard calibration guarantee. By instantiating this framework with different $\beta$'s, we have the flexibility of defining question-and-answer groups across which we would like calibration guarantees to hold. In this paper, we present an instantiation of $\beta$ as a kd-tree, an adaptive multivariate histogram method that provides cohesive groupings of QA pairs.

2. We propose two posthoc calibration techniques for QA-calibration: (a) QA binning and (b) scaling QA binning. The latter uses the former as a subroutine. This is particularly useful in the practical setting where some QA induced groups may lack data. We show that both methods satisfy a distribution-free approximate QA-calibration guarantee. In both cases, the approximation level is used to decide on the number of points per bin, a key hyperparameter.

3. Finally, we experiment on newly developed elicitation prompts across 5 commonly used QA benchmark datasets of various sizes. We find that QA binning and scaling QA binning achieve lower QA-calibration error than elicited confidence scores and other baselines. We further justify our framework by demonstrating its performance in a downstream selective QA task. Our posthoc calibration algorithms achieve around 10-40% increase in QA-calibration performance and up to 30% increase in selective answering performance.

---

[2]Recent research indicates that, even in a setting where one has access to token-based likelihood, it does not necessarily capture the overall semantic uncertainty (Kuhn et al., 2023).

| Question $q$ | Answer $a$ | Confidence $h(q,a)$ | $\mathbb{P}(Y=1|q,a)$ |
|:---:|:---:|:---:|:---:|
| $q_{11}$ | $a_{11}$ | 0.8 | 0.6 |
| $q_{12}$ | $a_{12}$ | 0.8 | 0.7 |
| $q_{21}$ | $a_{21}$ | 0.8 | 0.9 |
| $q_{22}$ | $a_{22}$ | 0.8 | 1.0 |

Table 1: QA pairs with confidence and correctness for Example 2.1.

## 2 DEFINING QA-CALIBRATION

### 2.1 NOTATION

We first define the question-and-answering process with verbalized confidence elicitation. In the most minimal form of interaction, a question $q$, is embedded into a prompt using a prompt function $m(q)$. The user obtains an answer $a = c(m(q))$ from an answering function $c : \mathcal{Q} \to \mathcal{A}$. Note that $c$ could be a randomized function, as typical LMs are. Confidence elicitation (Xiong et al., 2024; Tian et al., 2023) enhances the prompt to let the user obtain confidence $h(m(q), a)$ from the confidence function $h : \mathcal{Q} \times \mathcal{A} \to [0, 1]$. Note that both answering and confidence functions are implicit in the LM interaction. In the following, for simplicity, we omit the dependence on prompt function $m$, and use $c(q)$ for $c(m(q))$ and $h(q, a)$ for $h(m(q), a)$. We assume that we have access to the binary ground truth $y$ for each pair $(q, a)$, which indicates whether the answer $a$ is correct ($y = 1$) for the question $q$. Our $N$-element dataset is then $D = \{(q_i, a_i, h_i, y_i)\}_{i \in [N]}$ where $q_i$ is the $i$th question, $a_i = c(q_i)$, $h_i = h(q_i, a_i)$ and $y_i$ is the label which indicates whether the answer $a_i$ is correct for the question $q_i$. For more details, refer to Table 3. We assume that each instance $(q, a, h, y)$ of $D$ is an i.i.d. realization of r.v. $(Q, A = c(Q), H = h(Q, A), Y)$ drawn from a fixed distribution $P$ over the $\mathcal{Q} \times \mathcal{A} \times [0, 1] \times [0, 1]$ where

$$P := P_Q \times P_{A|q} \times P_{H|q,a} \times P_{Y|q,a} . \tag{1}$$

**Definition 2.1** (Calibration). We say that $h$ is calibrated for distribution $P$ if:

$$\mathbb{E}\left[Y \mid h(Q, A) = p\right] = p, \text{ a.s. for all } p \in [0, 1]^3 \tag{2}$$

In words, that the conditional distribution on $Y$ conditional on the prediction that $h(Q, A) = p$ is a Bernoulli distribution with bias $p$. In the following, we refer to Eq. ( 2) as average-case calibration to distinguish it from QA-calibration.

While perfect calibration is impossible in finite samples, a standard measure for calibration error of $h$ is defined as follows.

**Definition 2.2** (Expected (Average-case) Calibration Error). The expected (average-case) calibration error of $h$ is defined as:

$$\text{CE}(h) = \mathbb{E}_{Q,A}\left[|\mathbb{E}[Y \mid h(Q, A)] - h(Q, A)|\right] \tag{3}$$

### 2.2 QA-CALIBRATION INSTEAD OF STANDARD CALIBRATION

In practice, a user who obtains an answer $a$ to their question $q$ with confidence $p$ wants to know the probability that $y = 1$ among "similar" $(q, a)$ pairs with confidence $p$ (e.g., within the same topic of interest). A perfectly calibrated $h$ however, may not satisfy this requirement, as we illustrate in the following example[4]:

**Example 2.1.** Suppose the dataset consists of two sets of QA pairs, $\beta_1 = \{(q_{11}, a_{11}), (q_{12}, a_{12})\}$, and $\beta_2 = \{(q_{21}, a_{21}), (q_{22}, a_{22})\}$ with confidence function $h$ and $\mathbb{P}(Y = 1|q, a)$ as shown in Table 1.

It can easily be seen that $h$ is perfectly calibrated according to Eq. ( 2) when we assume that each QA pair have the same sampling probability. However, when we consider the sets $\beta_1$ and $\beta_2$ separately, each set is not well-calibrated.

$$\mathbb{E}[Y|h(q,a) = 0.8, \beta_1] = \frac{1}{2}(\mathbb{P}[Y = 1 \mid q_{11}, a_{11}] + \mathbb{P}[Y = 1 \mid q_{12}, a_{12}]) = \frac{1}{2}(0.6 + 0.7) = 0.65,$$

$$\mathbb{E}[Y|h(q,a) = 0.8, \beta_2] = \frac{1}{2}(\mathbb{P}[Y = 1 \mid q_{21}, a_{21}] + \mathbb{P}[Y = 1 \mid q_{22}, a_{22}]) = \frac{1}{2}(0.9 + 1.0) = 0.95.$$

---

[3]As $Y$ is a Bernoulli r.v., the LHS can also be written as $\Pr[Y = 1 \mid h(Q, A) = p]$.

[4]We give the smallest example where each group has more than one pair, since we are illustrating group calibration, but the argument holds when there is only one element per group.

Practically, when a user is only concerned with a particular set $\beta_1$ or $\beta_2$, *the calibration claim does not align with reality*. For $\beta_1$, $h$ is overconfident, and underconfident for $\beta_2$.

In the above, the expectation $\mathbb{E}[Y \mid h(q, a)]$ is not interpretable from a decision-making perspective. The space of generative QA pairs $\mathcal{Q} \times \mathcal{A}$ is intuitively too large for a user, particularly since they are typically only interested in a smaller subset of $(q, a)$ pairs. We hence consider a generalization of Eq. ( 2) necessary to perform adequate calibration for generative QA.

**Definition 2.3** (QA-calibration). $h$ is QA calibrated for the distribution $P$ in Eq. ( 1) if:

$$\mathbb{E}\left[Y \mid h(Q, A) = p, \beta(Q, A)\right] = p, \tag{4}$$

a.s. for all $p \in [0, 1]$, where $\beta : \mathcal{Q} \times \mathcal{A} \to \mathcal{S}$ is a fixed mapping to some embedding space $\mathcal{S}$.

In this work, we focus on a finite set $\mathcal{S}$ and $\beta$ is a (deterministic) discretization scheme that maps $(q, a)$ to a finite set $S$, i.e., $\beta$ induces a partitioning of the QA space according to the output of $\beta$. Note that this QA-calibration reduces to calibration in Eq. ( 2) if $\beta(q, a)$ is the same for all question-and-answers $(q, a)$. Intuitively, $\beta$ is chosen such that the pre-image of a specific value of $\beta$ represents a grouping that an end-user might be interested in.

Going back to Example 2.1, QA-calibration would tell us the following: "among question and answer pairs with confidence $p$ that are mapped to the same $\beta$ value (as the user's $(q, a)$ pair), the fraction who are correct ($y = 1$) is also $p$". Further, we also extend Eq. ( 3) to propose an error metric for QA-calibration:

**Definition 2.4** (QA-calibration error). The QA-calibration error of $h$ with fixed mapping $\beta$ is defined as:

$$\text{CE}(h; \beta) = \mathbb{E}_{Q,A}\left[|\mathbb{E}[Y \mid h(Q, A), \beta(Q, A)] - h(Q, A)|\right].$$

## 2.3 Generalizing (Average-case) Calibration via kd-tree Instantiation of $\beta$

The definition of QA-calibration (Definition 2.3), requires a predefined $\beta$ mapping. While our schemes developed in Section 3 can work with any arbitrary $\beta$ mapping, we propose a general approach for $\beta$ in the language model QA setting: the embed-then-bin method. This involves computing vector embeddings from question-and-answer pairs and then creating a multidimensional histogram over these embeddings. The range of $\beta$ is then set to index over the histogram bins of the embeddings. We write $\beta = \beta_{\text{bin}} \circ \beta_{\text{emb}}$, where $\beta_{\text{emb}} : \mathcal{Q} \times \mathcal{A} \to \mathbb{R}^M$ computes an $M$-dimensional embedding and $\beta_{\text{bin}} : \mathbb{R}^M \to \mathcal{S}$ computes the index of the embedding histogram bin containing the embedding.

Specifically, we use the `[CLS]` token embedding from the pre-trained DistilBERT model (Sanh et al., 2019): $\beta_{\text{emb}} : \mathcal{Q} \times \mathcal{A} \to \mathbb{R}^{768}$ ($M = 768$). As for the histogram, we adapt a kd-tree (Bentley, 1975) to bin each vector to an integer representing the index of the partition containing the vector: $\beta_{\text{bin}} : \mathbb{R}^{768} \to S$, where $S$ is the set of partition indices. See Appendix C for details on our kd-tree construction. An important hyperparameter is the maximum depth of the tree $d$, which determines the number of partitions.

While DistilBERT is designed to be smaller and faster than its alternatives, it is still able to generate high-quality contextual embeddings that are rich in semantic information. Next, a kd-tree performs efficient and adaptive binning for the high-dimensional embedding space by successively splitting along different dimensions. Each partition of the tree contains semantically-similar $(q, a)$ pairs that are "near" each other in the embedding space.

The choice of kd-tree generalizes the standard calibration as when $d = 0$, $\text{CE}(h; \beta)$ reduces to $\text{CE}(h)$. The hyperparameter $d$ should be chosen based on the downstream metric that needs to be optimized. In the experiments section, we further describe how $\beta_{\text{bin}}$ are constructed.

## 3 Achieving Posthoc QA-calibration

As discussed earlier, an elicited confidence score function ($h$) of LM-model is not guaranteed to be QA calibrated (or even calibrated under average-case). For any $\beta : \mathcal{Q} \times \mathcal{A} \to \mathcal{S}$, the goal of posthoc QA-calibration is to design a scheme that that can take any $h$ and transform it to a QA calibrated confidence score. We wish to learn a posthoc calibrator $g : [0, 1] \to [0, 1]$ such that $g \circ h$ is (approximately) QA calibrated using a calibration dataset $D = \left\{(q_i, a_i, h_i, y_i)\right\}_{i \in [N]}$. Additional missing details from this section are collected in Appendix A.

---



Algorithm 1: QA binning: Train-time Subroutine



---

**Input:** Calibration data: $\mathcal{D} = \{(q_i, a_i, h_i, t_i)\}_{i \in [n]}$. Hyperparameters: Minimum number of points per bin $b \in \mathbb{N}$ (say 50), tie-breaking parameter $\delta > 0$ (default: $10^{-10}$)
**Output:** Hash table $\mathcal{G}$ of fitted UMD calibrators $g_{\text{UMD}}$'s, keyed by partition index $s \in \mathcal{S}$
    */* First construct UMD calibrator for points outside of bounded kd-tree spaces and store in $\mathcal{G}$ with a default key, say* `root`*/*
  1:  $g_{\mathcal{D}} \leftarrow \text{UMD}(\mathcal{D}, B = \lfloor |\mathcal{D}|/b \rfloor, \delta = \delta)$.
  2:  $\mathcal{G}[\texttt{root}] \leftarrow g_{\mathcal{D}}$
  3:  **for** $s \in \mathcal{S}$ **do**
  4:     $\mathcal{D}_s \leftarrow \{(h_i, t_i) : \beta(q_i, a_i) = s\}_{i \in [n]}$
  5:     */* If $\mathcal{D}_s$ is empty then continue to the next $s$ */*
  6:     $n_s \leftarrow |\mathcal{D}_s|$
  7:     $\mathcal{G}[s] \leftarrow \text{UMD}(\mathcal{D} = \mathcal{D}_s, B = \lfloor n_s/b \rfloor, \delta = \delta)$
  8:  **end for**
  9:  **return** $\mathcal{G}$

---

Our recalibration schemes utilize the existing building blocks in posthoc calibration literature, like histogram binning (Zadrozny & Elkan, 2001) and scaling (Platt, 1999). The novelty of our recalibration methods lies how we generalize and combine these building blocks achieving the best of both worlds: our binning component generalizes histogram binning to any partitioning and enables distribution-free guarantees, and our hierarchical scaling component reduces overfitting, enabling learning to be performed across partitions which may have varied number of data points.

Our schemes take the $\beta$ function as input, and all our schemes and theoretical results hold for any $\beta$. In our experiments, we will use the kd-trees based $\beta$ instantiation from Section 2.3. We focus on methods that output discretized scores, since this has been shown to be easier to assess (Appendix A.1). We achieve this by first discretizing the QA space using $\beta$ and then ensuring that for each QA partition $s \in \mathcal{S}$ we output a calibrated confidence score. We start by describing an adaptation of the classical histogram binning (Zadrozny & Elkan, 2001) idea for achieving QA-calibration. In Section 3.2, we build upon this to provide a more robust algorithm that also uses scaling before binning.

## 3.1 Approach 1: QA binning

QA binning is a standalone posthoc QA-calibration algorithm. It uses as a subroutine, uniform-mass-double-dipping histogram binning (UMD) (Gupta & Ramdas, 2021), which we describe in Algorithm 4. Informally, the UMD procedure partitions the interval $[0, 1]$ into $B$ bins using the histogram of $h$ values from the calibration dataset $D$, ensuring that each bin has approximately the same number of calibration points. It returns a calibrator function $g_{\text{UMD}}$ that takes as input an uncalibrated confidence score, then allocates it to one of the $B$ bins, and returns the probability of label being 1 (estimated as the average of the $y$ values from the calibration dataset $D$ that are mapped into that bin).

In Algorithm 1, we present a subroutine that uses the UMD procedure to construct a different calibrator for each QA-partition, which we will invoke with different inputs. Algorithm 1 takes as input a dataset $\mathcal{D} = \{(q_i, a_i, h_i, t_i)\}_{i \in [n]}$ where $h_i$ is the (elicited) confidence score for the answer $a_i$ and $t_i$ is the (potentially noisy) label of the correctness of answer $a_i$ for the question $q_i$. Algorithm 1 partitions the input dataset $\mathcal{D}$ to $\{\mathcal{D}_s\}_{s \in \mathcal{S}}$, where $\mathcal{D}_s = \{(q_i, a_i, h_i, t_i) : \beta(q_i, a_i) = s\}$. For each $s \in \mathcal{S}$, UMD is fit using $\mathcal{D}_s$ to construct a $g_{\text{UMD}}$ calibrator that is stored in a hash table $\mathcal{G}$, keyed by $s$. We next describe the training and test processes of QA binning.

**Training Process.** We invoke Algorithm 1 on our calibration dataset $D$, i.e., $\mathcal{D} = D$. Hence, $t_i$ in Algorithm 1 is set to the ground truth $y_i$. The output is a hash table of UMD calibrator functions indexed by the entries in the range $S$ of $\beta$.

**Test Process.** For a test input $(q_{\text{test}}, a_{\text{test}}, h_{\text{test}})$, we invoke Algorithm 2. A fitted UMD ($g_{\text{UMD}}$) for this $(q_{\text{test}}, a_{\text{test}})$ pair is retrieved from the hash table $\mathcal{G}$ using $\beta(q, a)$ as key, which is then invoked to obtain a QA calibrated confidence score $g_{\text{UMD}}(h_{\text{test}})$. In our kd-tree instantation, if $\mathcal{G}$ does not have a calibrator for a $\beta(q_{\text{test}}, a_{\text{test}})$, we use UMD to calibrate that pair. This corresponds to points that lie outside of the bounded kd-tree spaces (refer to Appendix C). Since we can generate a large amount of question-and-answer pairs without requiring ground truth, the number of test points in our experiments that fall outside the bounded spaces is negligible.

---

Algorithm 2: QA binning: Test-time Subroutine

---

**Input:** Test question, answer and confidence score: $(q_{\text{test}}, a_{\text{test}}, h_{\text{test}})$, Hash table $\mathcal{G}$
**Output:** (Approximately) QA calibrated confidence score
  1: **if** $\beta(q_{\text{test}}, a_{\text{test}}) \in \mathcal{G}$ **then**
  2:     $g_{\text{UMD}} \leftarrow \mathcal{G}[\beta(q_{\text{test}}, a_{\text{test}})]$
  3:     **return** $g_{\text{UMD}}(h_{\text{test}})$
  4: **else**
  5:     /* In this case, the test input $(q_{test}, a_{test})$ does not lie in any of the bounded kd-tree spaces*/
  6:     $g_{\mathcal{D}} \leftarrow \mathcal{G}[\text{root}]$
  7:     **return** $g_{\mathcal{D}}(h_{\text{test}})$
  8: **end if**

---

Note that while we assumed here access to the true labels $y_i$'s, we can also operate QA binning with proxy labels generated by say another LM on QA pairs, as discussed more in following subsections. In practice, users are likely to define large $S$ (for example, large maximum depth of tree $d$) in order to obtain more cohesive groups of QA pairs. UMD, and therefore QA binning, may overfit if the number of data points within $\mathcal{D}_s$ is small — a highly likely scenario if $\beta$ induces very fine partitions. We overcome these issues with our next approach that combines scaling with binning.

## 3.2 APPROACH 2: SCALING QA BINNING

Scaling QA binning is a standalone posthoc QA-calibration algorithm, that is based on performing a scaling step prior to QA binning. In the original scaling-binning approach (Kumar et al., 2019), confidence scores are first scaled to their (maximum likelihood) fitted values using logistic regression, and the fitted values are then used as proxy ground truth for histogram binning. Adapting this paradigm, in our scaling QA binning, we use a scaling subroutine (defined below) to produce fitted values, which are then used as proxy ground truth for QA binning (e.g., $t$ in Algorithm 1 is set to these fitted values).

We adapt the scaling procedure from Kumar et al. (2019) with a crucial change. The confidence score distributions in different partitions $s \in \mathcal{S}$ may have different miscalibration profiles, since an LM may be underconfident for a partition but overconfident for another. While scaling is useful for average-case recalibration (Platt, 1999), for QA-calibration, we propose to use a hierarchical logistic regression model with partial pooling (Goldstein, 2011). Hierarchical model distinguishes between fixed effects (consistent across different partitions) and random effects (allowing for variations at different partitions). The latter explains the variability between partitions that may not be captured by fixed effects alone. Partial pooling allows for information sharing across partitions, which improves estimates for partitions with few data points. In our kd-tree instantiation, as the maximum depth parameter $d$ increases, some partitions may have only a few data points.

We next describe the training and test processes of scaling QA binning.

**Training Process.** With an abuse of notation, let $s[i]$ denote the partition index containing $(q_i, a_i)$. We define the scaler $g_{\text{scaler}} : [0, 1] \rightarrow [0, 1]$ via the following likelihood of $y_i$:

$$Y_i \sim \text{Bernoulli}\left(\text{logit}^{-1}\left(B_0 + U_{s[i]} + (B_1 + V_{s[i]})h_i\right)\right), \quad (5)$$

where $B_0$ is a fixed intercept, $U_{s[i]}$ is the random intercept for partition $s[i]$, $B_1$ is the fixed slope for confidence score $h_i$, and $V_{s[i]}$ is the random slope for partition $s[i]$. On top of random intercepts, which assumes a different additive baseline per partition, we use random slopes which allow the relationship between the accuracy and confidence score to differ for each partition.

We first split the calibration dataset $D$ (by default equally) into 2 sets, $D^1$ and $D^2$, and invoke Algorithm 3 on $D^1$ and $D^2$. Similar to the QA binning approach (Algorithm 1), this produces a hash table $\mathcal{G}$ of $g_{\text{UMD}}$'s. The partition of each instance $s$ determines which random intercept $U_s$ and slope $V_s$ to use. In our kd-tree instantiation, if a point lies outside of bounded kd-tree spaces, we set the random effects to zero, thus assuming that this point behaves similarly to the overall population average.

**Test Process.** Similar to QA binning, for a test input $(q_{\text{test}}, a_{\text{test}}, h_{\text{test}})$, we invoke Algorithm 2.

Due to the scaling step, the computational cost of scaling QA binning training process is more expensive than that of QA binning. Both schemes have the same test process.

---

Algorithm 3: Scaling QA binning: Train-time Subroutine

---

**Input:** Calibration data for scaling: $\mathcal{D}^1 = \{(q_i^1, a_i^1, h_i^1, t_i^1)\}_{i \in [n_1]}$. Calibration data for binning: $\mathcal{D}^2 = \{(q_i^2, a_i^2, h_i^2, t_i^2)\}_{i \in [n_2]}$. Hyperparameters: Minimum number of points per bin $b \in \mathbb{N}$ (say 50), tie-breaking parameter $\delta > 0$ (default: $10^{-10}$)
**Output:** Hash table $\mathcal{G}$ of fitted UMD calibrators $g_{\mathrm{UMD}}$'s, keyed by partition index $s \in \mathcal{S}$

1: $g_{\mathrm{scaler}} \leftarrow$ Fit Eq. ( 5) on $\{(h_i^1, t_i^1)\}_{i \in [n_1]}$
2: /* Construct proxy ground truth $\tilde{y}$. See Definition 3.2./
3: $\{\tilde{y}_i^2\}_{i \in [n_2]} \leftarrow \{g_{\mathrm{scaler}}(h_i^2)\}_{i \in [n_2]}$
4: $\mathcal{G} \leftarrow$ Execute Algorithm 1 with parameters $\mathcal{D} = \{(q_i^2, a_i^2, h_i^2, \tilde{y}_i^2)\}_{i \in [n_2]}, b = b, \delta = \delta$
5: **return** $\mathcal{G}$

---

## 3.3 DISTRIBUTION FREE ANALYSIS OF QA BINNING AND SCALING QA BINNING

In the next result, we prove a high probability bound on the QA-calibration error (Definition 2.4) of the above schemes. To formalize the guarantee, we adapt the conditional calibration notion from Gupta et al. (2020); Gupta & Ramdas (2021) to the QA-calibration setting, defined below.[5]

**Definition 3.1** (Conditional QA-calibration). Let $\epsilon, \alpha \in (0, 1)$ be some given levels of approximation and failure respectively. Confidence function $h : \mathcal{Q} \times \mathcal{A} \to [0, 1]$ is $(\epsilon, \alpha)$-conditionally QA calibrated for discretization scheme $\beta : \mathcal{Q} \times \mathcal{A} \to S$ if for every distribution $P$ defined in Eq. ( 1),

$$\mathbb{P}\left(\forall s \in \mathcal{S}, p \in \mathrm{range}(h), |\mathbb{E}[Y \mid h(Q, A) = p, \beta(Q, A) = s] - p| \leq \epsilon\right) \geq 1 - \alpha.$$

This is a distribution-free (DF) guarantee since they are required to hold for all distributions $P$ over $\mathcal{Q} \times \mathcal{A} \times [0, 1] \times \{0, 1\}$ without restriction. In order to estimate $\epsilon$ for both our posthoc algorithms, we permit label misspecification defined as follows:

**Definition 3.2.** (Misspecified proxy ground truth) Let the random variable $\tilde{Y} \in [0, 1]$ with distribution $P_{\tilde{Y}|(q,a)}$ be a proxy for ground truth $Y$. We constrain the misspecification in the following way: assume that there is some (minimal) $\nu \in [0, 1]$ such that for all $p \in [0, 1]$,

$$\max(\mathbb{E}[Y \mid h(Q, A) = p] - \nu, 0) \leq \mathbb{E}[\tilde{Y} \mid h(Q, A) = p] \leq \min(\mathbb{E}[Y \mid h(Q, A) = p] + \nu, 1),$$

In practice, $\tilde{y}$ can come from two sources: 1) an LM-constructed ground truth (see ground truth proxy in Table 3), and 2) the fitted values from our S step in scaling QA binning. Let $\tilde{D} = \{(q_i, a_i, h_i, \tilde{y}_i)\}_{i \in [N]}$, denote a (proxy) dataset with samples from $P = P_Q \times P_{A|q} \times P_{H|q,a} \times P_{\tilde{Y}|q,a}$. To prove calibration guarantees for our schemes, we rely on the following result.

**Theorem 3.1** (Distribution-free QA-calibration Guarantee). Consider an input calibration dataset $\tilde{D}$ defined above with misspecification factor $\nu$ from Definition 3.2. Assume that the $h_i$'s are distinct, number of points per bin $b \geq 2$, and number of instances within each partition $n_s \geq b$ for every $s \in \mathcal{S}$. The calibrator $g_{\mathrm{UMD}}$ retrieved in Line 2 of Algorithm 2, trained using Algorithm 1 with input $\mathcal{D} = \tilde{D}$, is $(\epsilon, \alpha)$-conditionally QA calibrated for any $\alpha \in (0, 1)$, with $\epsilon = \sqrt{\frac{\log(2N/b\alpha)}{2(b-1)}} + \nu$.

The proof is in Appendix A.3. The dependence on $\approx 1/\sqrt{b}$ factor comes because the Algorithm 1 delegates at least $b$ points to every bin. We now discuss how Theorem 3.1 is applicable for both QA binning and scaling QA binning with different $\nu$'s.

**Applying Theorem 3.1 to QA Binning & Scaling QA Binning.** In our description of QA binning (Section 3.1), we assumed $t$ is set to the ground truth $y$ (in Algorithm 1), hence, by definition $\nu = 0$. Theorem 3.1 can also be used to choose $b$, see the plots for $\nu = 0$ in Figure 2.

If the true labels are not available, then we can still use QA binning say by using an LM to produce proxy ground truth. In this case, the misspecification constant $\nu$ depends on the data generating process of misspecified labels. When an LM is used to produce proxy ground truth, if there is a hold-out set containing the ground truth, then a bound on $\nu$ can be estimated empirically.

In scaling QA binning, where $\tilde{y}$ is set to be the fitted values of a hierarchical logistic regression model, the magnitude of misspecification factor $\nu$ depends on the goodness-of-fit of the fitted values. In

---

[5]One could define $(\epsilon, \alpha)$-marginal QA-calibration: $\mathbb{P}\left(|\mathbb{E}[Y \mid h(Q, A), \beta(Q, A)] - h(Q, A)| \leq \epsilon\right) \geq 1 - \alpha$. Conditional calibration is a stronger definition than marginal, as it requires the deviation between $\mathbb{E}[Y \mid h(Q, A), \beta(Q, A)]$ and $h(Q, A)$ to be at most $\epsilon$ for every $(s, r)$, including rare ones, not just on average.

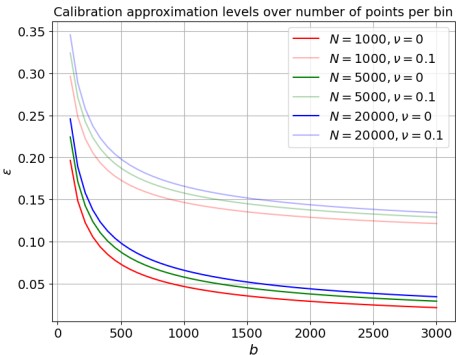

Figure 2: The relationship between $\epsilon$, number of points per bin $b$ and misspecification constant $\nu$ in Theorem 3.1. Based on the plot, when $\nu = 0$, practitioners should set $b \simeq 300$ when $N = 1000$, $b \simeq 400$ when $N = 5000$, $b \simeq 500$ when $N = 20000$ (attaining $\epsilon = 0.1$). When a ground truth proxy is misspecified (Definition 3.2), e.g., $\nu = 0.1$, for certain levels of $\epsilon$, the same bound can be attained with a larger $b$. For example, for achieving the same $\epsilon = 0.15$, if $\nu = 0$ then $b$ needs to be only approximately 250, whereas if $\nu = 0.1$ then $b$ has to be $> 1000$.

practice, we can estimate $\nu$ empirically using a hold-out dataset. This estimate can be used to choose $b$. For some levels of $\epsilon$, the same $\epsilon$ as in the case of $\nu = 0$ can be attained by setting $b$ to be a higher number. In our kd-tree instantiation, this amounts to using a smaller maximum depth hyperparameter.

## 4 RELATED WORK

**Calibration for Language Models.** Reinforcement learning from human feedback objective may prioritize adherence to user instructions in dialogue over producing well-calibrated predictions. (Kadavath et al., 2022). Lin et al. (2022) introduced the concept of verbalized confidence that prompts LMs to express confidence directly, focusing on fine-tuning, instead of zero-shot verbalized confidence. Mielke et al. (2022) uses an external calibrator for a white-box large language model. Other methods use consistency measures to improve LM calibration (Lyu et al., 2024). Our experimental setup closely relates to recent works in LM confidence elicitation (Tian et al., 2023; Xiong et al., 2024). These methods lack novel posthoc calibrators and do not offer the rigorous calibration guarantees that ours provide. Calibration has been shown to impact selective QA performance Kamath et al. (2020), but they focus on uncertainty quantification and assumes that the LM allows access to the model likelihood.

**Group Notions of Calibration.** Previous works highlight the limitations of average-case calibration. Group-wise calibration, which uses predefined groupings (Kleinberg et al., 2017; Pleiss et al., 2017), has been adapted for language models (LMs). Li et al. (2024) train a model that approximates the precision-threshold curve for a given group by using few-shot samples to predict the LM's empirical precision at various confidence thresholds. Ulmer et al. (2024) train an auxiliary model using accuracy per group as target to predict an LM's confidence based on textual input and output. Detommaso et al. (2024) achieves multicalibration — simultaneous calibration across various intersecting groupings of the data. Our work complements multicalibration, and our methods could extend to this by adapting Algorithm 3 in Detommaso et al. (2024). Luo et al. (2022) measure calibration over a set of similar predictions, quantified by a kernel function on feature space. Again, the notions of calibrations and their guarantees are incomparable.

**Other Metrics for Measuring Calibration Error.** Brier score (Brier, 1950) measures the accuracy of probabilistic predictions but while it can be decomposed into calibration and refinement (Blattenberger & Lad, 1985), it doesn't directly assess calibration. As a result, a model with lower squared error may still be less well-calibrated. Maximum Calibration Error (MCE) examines the maximum miscalibration across confidence bins (Guo et al., 2017), but in a QA setting, it faces the same issues as calibration error ($CE(h)$), as shown in Example 2.1. We generalize MCE to the QA-calibration setting in Appendix D. Through QA-calibration, we present a principled and interpretable calibration target for QA settings.

## 5 EXPERIMENTS

**Datasets, Models, and Prompts.** We use 5 QA datasets: TriviaQA (Joshi et al., 2017), SciQ (Welbl et al., 2017), BigBench (Srivastava et al., 2022), OpenBookQA (Mihaylov et al., 2018), and MMLU (Hendrycks et al., 2021) (see Table 4 for more details). We use two performant models: Mistral (Jiang et al., 2023) and Gemma (Team et al., 2024). To elicit confidence scores, we use two prompt techniques recently suggested in literature: Verb1S-Top1 & Ling1S-Top1 from Tian et al. (2023). See Table 5, (Appendix B.1) for details about the prompts. Central to the implementation of posthoc calibration and evaluation of calibration is the availability of a label for a question-and-answer

pair — specifically, whether the answer provided by the LM is accurate for the question posed by the user. In practice, a common idea to generate this label is to take an LM provided *answer*, and then use another LM to assess whether the proposed answer is semantically equivalent to the true (ground truth) answer (Tian et al., 2023). To construct the ground truth proxy for $y$, we use Llama 3.1 (Dubey et al., 2024). See Appendix E for qualitative examples of question-and-answer pair kd-tree partition assignments.

**Our Methods.** We compare the performance of our calibrators: QA binning (QAB) from Subsection 3.1 and hierarchical scaling QA binning (HS-QAB) from Subsection 3.2. We also include a fully pooled version of scaling QA binning (S-QAB), by setting $s[i]$ to a constant (thus, one partition) in Eq. ( 5). To set the hyperparameter minimum number of points per bin $b$ (Algorithm 1), we set an $\epsilon$ that is not too large as per Figure 2 and use root finding with the $\epsilon$ expression in Theorem 3.1 to choose $b$. We then search over a range of $b$'s by allowing for a misspecification range between $0$ and $0.05$ and a range of maximum kd-tree depths depending on the size of the dataset such that each partition admits a 3–10 bins. To set $B$ in UMD, we follow the guidelines in Gupta & Ramdas (2021). Note that their bound does not involve misspecification factor.

**Baselines.** We consider the following baselines: no recalibration (None) which returns state-of-the-art elicited confidence scores (Tian et al., 2023), histogram binning (UMD) (Gupta & Ramdas, 2021), Platt scaling (S) (Platt, 1999), and scaling-binning (S-B) (Kumar et al., 2019). These baselines consist of the state-of-the-art ideas in posthoc calibration. Note that the techniques UMD, S, and S-B, aim to minimize the expected calibration error CE (Definition 2.2) and do not take the partitions induced by $\beta$ into account.

**Metrics.** In this section, we primarily use two metrics for comparison. First, is the QA-calibration error $\text{CE}(h; \beta)$ (Definition 2.4), which is the metric that our methods (presented in Section 3) optimize for. A lower $\text{CE}(h; \beta)$ indicates a more effective scheme for achieving QA-calibration. As explained in Section 2, QA-calibration error generalizes the expected calibration error ($\text{CE}(h)$). Second, to measure the downstream impact of using QA-calibration as the confidence calibration notion in a QA setting, we adapt selective question answering (Kamath et al., 2020) to our setting: given a threshold $\gamma \in [0, 1]$, the answer $a$ is returned for a question $q$ to the end user if $h(q, a) \geq \tau$, and abstains (no answer is returned) otherwise.[6] We adapt the area under risk-coverage curve, which is a standard way to evaluate selective prediction methods (El-Yaniv & Wiener, 2010) to our setting. Our posthoc schemes produce discretized confidence scores $h$, often with large number of ties, which lead to unreliable risk-coverage calculations. Therefore, we measure the area under the accuracy-confidence curve (AUAC) by sampling grid points from 0 to 1, computing the accuracy (based on ground truth $y$) of points with confidence greater than each grid point. A higher AUAC indicates a more effective selective QA scheme.

**Training.** We perform a 4-way (20:60:10:10) split of each dataset: the first is used to construct the kd-tree, second is used for posthoc calibration training, third is used for hyperparameter tuning and fourth is for testing. We find it crucial to optimize for AUAC during hyperparameter tuning as our schemes already aim to minimize $\text{CE}(h; \beta)$, obtaining the appropriate maximum kd-tree depth $d$ and binning parameters $b$ and $B$. Missing experimental details are presented in Appendix B.1.

**Results.** Table 2 shows the performance of the posthoc calibrators on MMLU and BigBench datasets. More results are provided in Tables 6 and 7. Our methods (QAB, HS-QAB, and S-QAB) generally achieve the best QA-calibration error $\text{CE}(h; \beta)$ and best area under the accuracy-confidence curve (AUAC). While the first result, in itself, may not be surprising as our proposed schemes aim to minimize $\text{CE}(h; \beta)$, the gap between our techniques and baselines is significantly huge. For example, notice the difference in the calibration error when just using the SOTA confidence elicitation prompts (None) from (Tian et al., 2023) vs. our schemes in Table 2. Among our schemes, HS-QAB generally performs best. This is because a parametric model (especially a partially pooled model like hierarchical scaling) helps reduce the variance of the downstream binning averages.

For selective QA, we again notice that our proposed schemes consistently outperforms the baselines. In some cases, the underlying LLM using confidence elicitation prompts (None) is performant in selective answering and attains high AUAC (similar results using different LLMs were noted by Tian et al. (2023)), but are not well-calibrated as demonstrated by their high QA-calibration score. Our schemes, S-QAB and HS-QAB are generally the top-two performing schemes for this task with comparable and in many cases better AUAC scores than the None scheme. Since the accuracy-confidence curve is generated by examining the accuracy of answers above a confidence threshold, the results demonstrate the desirable quality that the confidence scores provided by our proposed

---

[6]For example, when $\tau = 0$, all answers are returned irrespective of the confidence score.

| Dataset | Prompt | LLM | Calibrator | $CE(h; \beta)$ | AUAC |
|---|---|---|---|---|---|
| MMLU | Ling1s-Top1 | Mistral | QAB (**ours**) | $0.171 \pm 0.005$ | $0.20 \pm 0.023$ |
| MMLU | Ling1s-Top1 | Mistral | HS-QAB (**ours**) | $\mathbf{0.16 \pm 0.005}$ | $\mathbf{0.269 \pm 0.022}$ |
| MMLU | Ling1s-Top1 | Mistral | S-QAB (**ours**) | $0.163 \pm 0.003$ | $0.19 \pm 0.045$ |
| MMLU | Ling1s-Top1 | Mistral | S-B | $0.393 \pm 0.004$ | $0.141 \pm 0.003$ |
| MMLU | Ling1s-Top1 | Mistral | S | $0.249 \pm 0.005$ | $0.122 \pm 0.002$ |
| MMLU | Ling1s-Top1 | Mistral | B | $0.392 \pm 0.003$ | $0.139 \pm 0.007$ |
| MMLU | Ling1s-Top1 | Mistral | None | $0.532 \pm 0.008$ | $\mathbf{0.269 \pm 0.007}$ |
| MMLU | Ling1s-Top1 | Gemma | QAB (**ours**) | $0.232 \pm 0.005$ | $0.211 \pm 0.032$ |
| MMLU | Ling1s-Top1 | Gemma | HS-QAB (**ours**) | $0.182 \pm 0.005$ | $0.26 \pm 0.01$ |
| MMLU | Ling1s-Top1 | Gemma | S-QAB (**ours**) | $\mathbf{0.181 \pm 0.009}$ | $\mathbf{0.275 \pm 0.029}$ |
| MMLU | Ling1s-Top1 | Gemma | S-B | $0.382 \pm 0.006$ | $0.184 \pm 0.002$ |
| MMLU | Ling1s-Top1 | Gemma | S | $0.201 \pm 0.009$ | $0.19 \pm 0.003$ |
| MMLU | Ling1s-Top1 | Gemma | B | $0.385 \pm 0.003$ | $0.183 \pm 0.008$ |
| MMLU | Ling1s-Top1 | Gemma | None | $0.603 \pm 0.006$ | $0.249 \pm 0.01$ |
| MMLU | Verb1s-Top1 | Mistral | QAB (**ours**) | $0.197 \pm 0.004$ | $0.198 \pm 0.046$ |
| MMLU | Verb1s-Top1 | Mistral | HS-QAB (**ours**) | $\mathbf{0.149 \pm 0.004}$ | $\mathbf{0.306 \pm 0.035}$ |
| MMLU | Verb1s-Top1 | Mistral | S-QAB (**ours**) | $0.16 \pm 0.004$ | $0.217 \pm 0.057$ |
| MMLU | Verb1s-Top1 | Mistral | S-B | $0.362 \pm 0.005$ | $0.139 \pm 0.004$ |
| MMLU | Verb1s-Top1 | Mistral | S | $0.258 \pm 0.005$ | $0.126 \pm 0.001$ |
| MMLU | Verb1s-Top1 | Mistral | B | $0.352 \pm 0.004$ | $0.132 \pm 0.011$ |
| MMLU | Verb1s-Top1 | Mistral | None | $0.639 \pm 0.006$ | $0.255 \pm 0.008$ |
| MMLU | Verb1s-Top1 | Gemma | QAB (**ours**) | $0.197 \pm 0.006$ | $\mathbf{0.345 \pm 0.035}$ |
| MMLU | Verb1s-Top1 | Gemma | HS-QAB (**ours**) | $\mathbf{0.151 \pm 0.006}$ | $0.3 \pm 0.035$ |
| MMLU | Verb1s-Top1 | Gemma | S-QAB (**ours**) | $0.161 \pm 0.006$ | $0.227 \pm 0.074$ |
| MMLU | Verb1s-Top1 | Gemma | S-B | $0.361 \pm 0.007$ | $0.133 \pm 0.005$ |
| MMLU | Verb1s-Top1 | Gemma | S | $0.228 \pm 0.005$ | $0.137 \pm 0.004$ |
| MMLU | Verb1s-Top1 | Gemma | B | $0.352 \pm 0.004$ | $0.144 \pm 0.011$ |
| MMLU | Verb1s-Top1 | Gemma | None | $0.636 \pm 0.007$ | $0.258 \pm 0.006$ |

Table 2: Performance of our QA-calibration methods, QA binning (QAB), Pooled scaling QA binning (S-QAB) and Hierarchical scaling QA binning (HS-QAB)), compared to the baselines, UMD, (B, (Gupta & Ramdas, 2021)), Platt Scaling (S, (Platt, 1999)), and Scaling-binning (S-B, (Kumar et al., 2019)), and prompt-based approach None (Tian et al., 2023) . The bold entry in the columns for $CE(h; \beta)$ and AUAC represent the best performing schemes for that metric. However, note that in some case the confidence intervals overlap, especially between our schemes.

schemes are better at ranking accurate answers higher than inaccurate ones. Crucially, the lower performance of other baselines (S-B, S, and B) demonstrates the advantage of QA-calibration. In particular, our QA-calibration framework identifies the optimal kd-tree depth, which empirically is never equal to zero (standard average-case calibration). We observe similar results in Appendix F when $\beta$ is varied. Lastly, we analyze the shift in confidence scores after posthoc calibration in Appendix G and per-partition reliability in Appendix H.

## 6 CONCLUSIONS

We proposed QA-calibration, a new notion of calibration which conditions on groups of QA pairs. We propose two new posthoc calibration schemes for LM-elicited confidence scores. Our algorithms are effective on various QA datasets. For future work, we plan to investigate alternative notions of calibration for groups, such as multi-calibration.

**Limitations.** The interpretability of the calibration guarantee for the user largely depends on the choice of $\beta$ — if users want the partitions to be very fine- or coarse-grained, then $\beta$ must be built with the appropriate depth. Furthermore, our algorithms assume that the output space of $\beta$ is fixed, which may be a limited assumption, given that the "information" space of generative QA may increase indefinitely over time.

## ACKNOWLEDGEMENTS

P.M was funded in part by the Hybrid Intelligence Center, a 10-year programme funded by the Dutch Ministry of Education, Culture and Science through the Netherlands Organisation for Scientific Research with grant number 024.004.022.

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

| Name | Notation | Description | Example |
|---|---|---|---|
| Question | $q$ | Inputted question. | What happens to you if you eat watermelon seeds? |
| Ground truth | $a^{gt}$ | Intended output answer. Note that we omit this in our mathematical description and work directly with the ground truth label $y$. In practice we use $a^{gt}$ to construct the ground truth proxy. | The watermelon seeds pass through your digestive system |
| Prompting function | $m()$ | A function that converts the question into a specific form by inserting the question $q$. The example prompting function is Verb. 1S top-$k$ in Tian et al. (2023), with $k = 3$. We denote the filled (by the question) prompt as $m(q)$. Note that multiple prompts may be generated (see the 2S prompts in Tian et al. (2023)) | Provide your three best guesses and the probability that each is correct (0.0 to 1.0) for the following question. Give ONLY the guesses and probabilities, no other words or explanation. For example: G1: <first most likely guess, as short as possible; not a complete sentence, just the guess! >P1: <the probability between 0.0 and 1.0 that G1 is correct, without any extra commentary whatsoever; just the probability!>... G3: <third most likely guess, as short as possible; not a complete sentence, just the guess!>P3: <the probability between 0.0 and 1.0 that G3 is correct, without any extra commentary whatsoever; just the probability!>. The question is: <$q$> |
| Answering function | $c(q)$ | An answering function $c(q) = c(m(q))$, that models the pipeline, that takes as input $m(q)$ and outputs an answer $a$. This is invoked $k$ times to obtain $k$ answers. The function subsumes the postprocessing performed on the LM response to obtain answer $a$. | Implicitly-defined in the LM interaction |
| Answers | $(a^1, a^2, a^3)$ | Three sampled answers, obtained after text normalization of the LLM raw output. | (nothing, grow watermelon, stomachache) |
| Confidence function | $h(q, a)$ | A confidence function $h(q, a) = h(m(q), a)$, that models the pipeline, that takes as input $m(q)$ and answer $a$ and outputs the confidence that the answer $a$ is correct for question $q$. This is invoked $k$ times to obtain $k$ confidences. The function subsumes the postprocessing performed on the LM response to obtain a float confidence value. | Implicitly-defined in the LM interaction |
| Confidence values | $(h^1, h^2, h^3)$ | Three confidence values associated with the three answers. Note that they may not be normalized. | (0.95, 0,05, 0.2) |
| Ground truth proxy (proxy of $y$) | $g(q, a, a^{gt})$ | The returned truth value is postprocessed to map Yes/No to 1/0. In practice, this is used as a proxy ground truth. We also experimented with the query (checking for semantic equivalence) from Tian et al. (2023), but we observe a high false negative rate. | We query an LLM using the following prompt: Do following two answers to my question Q agree with each other? Q: <$q$>, A1: <$a^1$>, A2: <$a^2$>. Please answer with a single word, either "Yes." or "No.", and explain your reasoning. |

Table 3: Important variables. Note that in our mathematical description, we treat the $k = 3$ output answers and confidences as 3 binary instances. Given a prompt function $m()$ and question $q$, at the end of an LM's interaction, we obtain answers and confidences.

# A  ADDITIONAL DETAILS FOR SECTION 3

## A.1  LIMITATIONS OF NON-DISCRETIZED METHODS

Estimation of $\mathbb{E}[Y|H]$ when $H = h(Q, A)$ is continuous is a difficult task (Kumar et al., 2019). The confidence must also be discretized for it to achieve guarantees of marginal calibration (Gupta & Ramdas, 2021). Prompts used to elicit confidence (Tian et al., 2023; Xiong et al., 2024) are not guaranteed to induce a discretized confidence score (both in the average-case- and QA-calibration cases).

In practice, the estimation of $CE(h; \beta)$ is done by partitioning $\mathcal{D}$ according to $\beta$: $\mathcal{D} = \cup_{s=1}^M \mathcal{D}_s$, where $\mathcal{D}_s = \{(q_i, a_i, h_i y_i) : \beta(q_i, a_i) = s\}$, and taking the weighted average of $CE(h)$'s (from Eq. ( 3)) from each $\mathcal{D}_s$ with weight $|\mathcal{D}_s|$. For each $\mathcal{D}_s$, $H$ is typically binned into intervals, and the calibration error in each bin is estimated as the difference between the average of confidence values and labels in that bin.

Lastly, estimation of $CE(h)$ for non-discretized methods may involve binning and can underestimate the error (Proposition 3.3 in (Kumar et al., 2019)). In our case, this is a possibility when comparing scaling against the other approaches.

## A.2  UNIFORM-MASS-DOUBLE-DIPPING HISTOGRAM BINNING (UMD)

We adapt UMD (Algorithm 1 from Gupta & Ramdas (2021)) to our notation in Algorithm 4. UMD takes as input calibration data $\mathcal{D} = \{(h_i, t_i)\}_{i \in [N]}$, where $h_i$'s are (uncalibrated) confidence scores and $t_i$'s are the corresponding target labels. The function order-stats returns ordered confidence scores $(h_{(1)}, \ldots, h_{(n)})$ where $h_{(1)} < h_{(2)} < \ldots < h_{(n)}$.

---

### Algorithm 4: UMD

**Input:** Number of bins $B$, Calibration data $\mathcal{D} = \{(h_i, t_i)\}_{i \in [N]}$ (dataset cardinality $n$ depends on the inputted dataset, we refer to the dataset within the algorithm as $\mathcal{D}$.)
**Output:** Calibrator function $g_{\text{UMD}} : [0, 1] \to [0, 1]$
1: $(h_{(1)}, \ldots, h_{(n)}) \leftarrow$ order-stats$(h_1, \ldots, h_n)$
2: $(t_{(1)}, \ldots, t_{(n)}) \leftarrow (t_1, \ldots, t_n)$ ordered as per the ordering of $(h_{(1)}, \ldots, h_{(n)})$
3: $\Delta \leftarrow \frac{(n+1)}{B}$
4: $\hat{\Pi} \leftarrow$ empty array of size $B$
5: $A \leftarrow$ 0-indexed array$([0, \lceil \Delta \rceil, \lceil 2\Delta \rceil), \ldots, n + 1]$
6: **for** $b \leftarrow 1$ to $B$ **do**
7: $\quad l \leftarrow A_{b-1}$
8: $\quad u \leftarrow A_b$
9: $\quad \hat{\Pi}_b \leftarrow$ mean$(t_{(l+1)} \ldots, t_{(u-1)})$
10: **end for**
11: $(h_{(0)}, h_{(n+1)}) \leftarrow (0, 1)$
12: Define the calibrator function, $g_{\text{UMD}}$: $g_{\text{UMD}}(h_{\text{test}}) = \sum_{b=1}^B \mathbf{1}\{h_{(A_{b-1})} \leq h_{\text{test}} < h_{(A_b)}\}\hat{\Pi}_b$
13: **return** $g_{\text{UMD}}$

---

## A.3  DISTRIBUTION-FREE GUARANTEES

In Gupta et al. (2020), UMD procedure (Algorithm 4) is assumed to take ground truth $y$ as input ($t = y$). Since in our setting, it is possible to pass a proxy ground truth, we now describe a generalization of conditional calibration guarantees of UMD procedure (Gupta & Ramdas, 2021, Theorem 3).

**Theorem A.1** (Conditional Calibration Guarantee of Algorithm 4 under Label Misspecification)**.**
Consider an input calibration dataset $\tilde{D}$ defined in Section 3.3 with misspecification factor $\nu$ from Definition 3.2. Assume that the $h_i$'s are distinct and $N \geq 2B$. Then calibrator $g_{\text{UMD}}$ outputted by Algorithm 4, with input $\mathcal{D} = \tilde{D}$, is $(\epsilon, \alpha)$-conditionally calibrated for any $\alpha \in (0, 1)$, with

$$\epsilon = \sqrt{\frac{\log(2B/\alpha)}{2(\lfloor N/B \rfloor - 1)}} + \nu. \tag{6}$$

*Proof.* For $b \in \{0, 1, \ldots, B\}$, define $k_b = \lceil b(N + 1/B) \rceil$. Fix $h_{(0)} := 0$ and $h_{(N+1)} := 1$ as the smallest and largest order statistics, respectively. As per Algorithm 4, we compute the order statistics of the input data and gather the following points into a set: $\mathcal{M} := \{h_{(k_1)}, \ldots, h_{k_{(B-1)}}\}$.

Let $\mathcal{B} : [0, 1] \rightarrow [B]$ be the binning function: $\mathcal{B}(H) = b \iff h_{(k_{b-1})} \le g_{\mathrm{UMD}}(H) < h_{(k_b)}$. Given $\mathcal{M}$, the function $\mathcal{B}$ is deterministic, i.e., for every $b \in [B]$, $\mathbb{E}[\tilde{Y}|\mathcal{B}(H) = b]$ is deterministic.

Consider some $b \in \{0, 1, \ldots, B\}$ and denote $l = k_{b-1}, u = k_b$. By Lemma 2 from (Gupta & Ramdas, 2021), the unordered (denoted by $h_{\{\cdot\}}$) confidence scores $h_{\{l+1\}}, h_{\{l+2\}}, \ldots, h_{\{u+1\}}$ are i.i.d given $\mathcal{M}$, with the same conditional distribution as that of $H$ given $\mathcal{B}(H) = b$. Therefore, $\tilde{y}_{\{l+1\}}, \tilde{y}_{\{l+2\}}, \ldots, \tilde{y}_{\{u-1\}}$ are i.i.d given $\mathcal{S}$, with the conditional distribution Bernoulli($\mathbb{E}[\tilde{Y} \mid \mathcal{B}(H) = b]$.

We now show that $\hat{\Pi}_b$ (defined in Line 9 and returned by $g_{\mathrm{UMD}}$ in Line 12 in Algorithm 4), the average of the $\tilde{y}$ in a bin $b$ values, concentrates around $\mathbb{E}[\tilde{Y} \mid \mathcal{B}(H) = b]$. For any $\gamma \in (0, 1)$, by Hoeffding's inequality, w.p. at least $1 - \gamma$:

$$\left| \mathbb{E}[\tilde{Y}|\mathcal{B}(H) = b] - \hat{\Pi}_b \right| \le \sqrt{\frac{\log(2/\gamma)}{2\lfloor u - l - 1 \rfloor}} \le \sqrt{\frac{\log(2/\gamma)}{2(\lfloor N/B \rfloor - 1)}}, \tag{7}$$

where the second inequality holds since for any $b$, $u - l = \lfloor (b+1)(N+1)/B \rfloor - \lfloor b(N+1)/B \rfloor \ge \lfloor N/B \rfloor$. Using law of total probability (partitioning $\{\mathcal{B}(h) = b\}$ to $\{h : \mathcal{B}(h) = b\}$), we obtain:

$$\mathbb{E}[\tilde{Y} \mid \mathcal{B}(H) = b] \le |\mathbb{E}[Y \mid \mathcal{B}(H) = b] - \nu|,$$

which implies that

$$\left| \mathbb{E}[Y \mid \mathcal{B}(H) = b] - \hat{\Pi}_b \right| \le \sqrt{\frac{\log(2/\gamma)}{2(\lfloor N/B \rfloor - 1)}} + \nu. \tag{8}$$

Set $\gamma = \alpha/B$ in Eq. (8), and take a union bound over all $b \in B$. With probability at least $1 - \alpha$, for every $b \in B$, $\left| \mathbb{E}[Y|\mathcal{B}(H) = b] - \hat{\Pi}_b \right| \le \epsilon$, where $\epsilon$ is the RHS of Eq. (8). This implies that:

$$\begin{aligned}
|\mathbb{E}[Y \mid H] - H| &= |\mathbb{E}[\mathbb{E}[Y \mid \mathcal{B}(H), H] \mid H] - H| \\
&= |\mathbb{E}[\mathbb{E}[Y \mid \mathcal{B}(H)] \mid H] - H| \\
&= |\mathbb{E}[\mathbb{E}[Y \mid \mathcal{B}(H)] - H \mid H]| \\
&= \left| \mathbb{E}[\mathbb{E}[Y \mid \mathcal{B}(H)] - \hat{\Pi}_{\mathcal{B}(H)} \mid H] \right| \\
&\le \mathbb{E}\left[ \left| \mathbb{E}[Y \mid \mathcal{B}(H)] - \hat{\Pi}_{\mathcal{B}(H)} \right| \mid H \right] \\
&\le \epsilon,
\end{aligned}$$

Where the first equality is due to the law of total expectation, the fourth equality is by the definition of the quantity returned by $g_{\mathrm{UMD}}$, and the first inequality is due to Jensen's inequality. This completes the proof, showing that $g_{\mathrm{UMD}}$ is $(\epsilon, \alpha)$-conditionally calibrated for any $\alpha \in (0, 1)$. $\qquad \square$

**Theorem 3.1** (Distribution-free QA-calibration Guarantee). Consider an input calibration dataset $\tilde{D}$ defined above with misspecification factor $\nu$ from Definition 3.2. Assume that the $h_i$'s are distinct, number of points per bin $b \ge 2$, and number of instances within each partition $n_s \ge b$ for every $s \in \mathcal{S}$. The calibrator $g_{\mathrm{UMD}}$ retrieved in Line 2 of Algorithm 2, trained using Algorithm 1 with input $\mathcal{D} = \tilde{D}$, is $(\epsilon, \alpha)$-conditionally QA calibrated for any $\alpha \in (0, 1)$, with $\epsilon = \sqrt{\frac{\log(2N/b\alpha)}{2(b-1)}} + \nu$.

*Proof.* For $s \in \mathcal{S}$, let $P_s$ denote the distribution of $(h, \tilde{y})$ conditional on $\beta(q, a) = s$. The tuples in $\mathcal{D}_s$ are i.i.d. samples from $P_s$ and $g_s = \mathcal{G}[s]$ is the corresponding fitted UMD calibrator. The number of bins is $B_s = \lfloor n_s/b \rfloor$. Since, $n_s \ge b\lfloor n_s/b \rfloor \ge 2B_s$,

Let $B = \sum_{s=1}^{S} B_s$ and $\alpha_s = \alpha B_s/B$. Note that $B \le \sum_{s \in \mathcal{S}} n_s/b = N/b$. We apply the bound in Theorem A.1 to obtain:

$$\mathbb{P}\left(\forall p \in \text{range}(g_s), \left|\mathbb{E}\left[\tilde{Y}|\beta(q,a)=s, g_s(h)=p\right]-p\right| \leq \sqrt{\frac{\log(2B_s/\alpha)}{2(\lfloor n_s/B_s\rfloor-1)}}+\nu \ \middle| \ \beta(q,a)=s\right)$$
$$\geq 1-\alpha_s.$$

Note that

$$\sqrt{\frac{\log(2B_s/\alpha_s)}{2(\lfloor n_s/B_s\rfloor-1)}}=\sqrt{\frac{\log(2B/\alpha)}{2(\lfloor n_s/B_s\rfloor-1)}}\leq\sqrt{\frac{\log(2N/b\alpha)}{2(\lfloor n_s/B_s\rfloor-1)}}\leq\sqrt{\frac{\log(2N/b\alpha)}{2(b-1)}}.$$

For every $s \in \mathcal{S}$:

$$\mathbb{P}\left(\forall p \in \text{range}(g_s), \left|\mathbb{E}\left[\tilde{Y}|\beta(q,a)=s, g_s(h)=p\right]-p\right| \leq \epsilon \ \middle| \ \beta(q,a)=s\right) \geq 1-\alpha_s.$$

Taking union bound over $S$ gives:

$$\mathbb{P}\left(\forall s \in \mathcal{S}, \forall p \in \text{range}(g_s), \left|\mathbb{E}\left[\tilde{Y}|\beta(q,a)=s, g_s(h)=p\right]-p\right| \leq \epsilon \ \middle| \ \beta(q,a)=s\right) \geq 1-\sum_{s\in\mathcal{S}}\alpha_s$$
$$= 1-\alpha.$$

This completes the proof. $\qquad\square$

To remove the assumption that $h_i$'s are distinct in Theorem 3.1, we could use the randomized version of UMD (Algorithm 2 in Gupta & Ramdas (2021)). This adds an additive factor to $\epsilon$ in Theorem 3.1 but can be made arbitrarily small. For simplicity, we have chosen to provide the guarantees for the non-randomized version of UMD.

## B ADDITIONAL DETAILS FOR SECTION 5

### B.1 MISSING EXPERIMENTAL DETAILS

**Details on LM Querying.** We set the LM `temperature` to close to 0 to minimize output stochasticity and set `max tokens` to be able to process the prompt $m(q)$. We include the prompts in Table 5. The variable EXPRESSION_LIST in Ling1S-Top1 prompt is taken from Fagen-Ulmschneider (2015).

**Compute Resources.** The experiments were run using a 3090Ti GPU and 64 GB of RAM.

**Training Details.**

We perform random splitting 8 times using different seeds and use the repeated measures to form our confidence intervals. We set the number of bins $B$ in the UMD baseline and scaling-binning by deriving hyperparameter search ranges from setting different $\epsilon$s in Theorem 3 in (Gupta & Ramdas, 2021). We describe how we set the number of points per bin $b$ in Algorithm 1 and Algorithm 3 in Section 5. We have also attempted to ensure that for every $B$ we can find in its hyperparameter tuning range, there is a $b$ in its hyperparameter tuning range, such that $B=\lfloor N/b\rfloor$. This correspondence, coupled with the same target variable to optimize, helps isolate the impact of $\beta$ on posthoc calibration.

| Dataset | Size | Type |
|---|---|---|
| SciQ | 11609 | QA |
| Bigbench | 20320 | QA |
| MMLU | 13869 | Multiple-choice QA |
| OpenBookQA | 4835 | Multiple-choice QA |
| TriviaQA | 11313 | QA |

Table 4: Dataset statistics. Multiple-choice QA datasets provide a set of possible answers. We add additional prompt text for them, see Table 5.

| Prompt Name | Prompt Template |
|---|---|
| Verb1S-Top1 | Provide your best guess and the probability that it is correct (0.0 to 1.0) for the following question. Give ONLY the guess and probability, no other words or explanation. For example: Guess: <most likely guess, as short as possible; not a complete sentence, just the guess! > Probability: <the probability between 0.0 and 1.0 that your guess is correct, without any extra commentary whatsoever; just the probability! > The question is: $<q>$. |
| Ling1S-Top1 | Provide your best guess for the following question, and describe how likely it is that your guess is correct as one of the following expressions: $EXPRESSION_LIST. Give ONLY the guess and your confidence, no other words or explanation. For example: Guess: <most likely guess, as short as possible; not a complete sentence, just the guess!> Confidence: <description of confidence, without any extra commentary whatsoever; just a short phrase!>  The question is: <q > |
| Additional prompt text for multiple-choice QA task with a set of choices $C$ | The answer must be chosen from the following list of size $<|C|>$: $<C>$. Only the actual answer (not the choice number or index) from the list should be used in the response. |
| ground truth proxy | See the example for ground truth Proxy in Table 3. |

Table 5: Prompts used in the experiments.

| Dataset | Prompt | LLM | Calibrator | CE$(h; \beta)$ | AUAC |
|---|---|---|---|---|---|
| SciQ | Ling1S-Top1 | Mistral | QAB (**ours**) | **0.165 ± 0.003** | **0.482 ± 0.024** |
| SciQ | Ling1S-Top1 | Mistral | HS-QAB (**ours**) | 0.175 ± 0.004 | 0.437 ± 0.028 |
| SciQ | Ling1S-Top1 | Mistral | S-QAB (**ours**) | 0.171 ± 0.006 | 0.426 ± 0.034 |
| SciQ | Ling1S-Top1 | Mistral | S-B | 0.466 ± 0.003 | 0.21 ± 0.007 |
| SciQ | Ling1S-Top1 | Mistral | S | 0.255 ± 0.005 | 0.21 ± 0.005 |
| SciQ | Ling1S-Top1 | Mistral | B | 0.457 ± 0.003 | 0.356 ± 0.025 |
| SciQ | Ling1S-Top1 | Mistral | None | 0.366 ± 0.011 | 0.451 ± 0.009 |
| SciQ | Ling1S-Top1 | Gemma | QAB (**ours**) | 0.158 ± 0.011 | 0.357 ± 0.023 |
| SciQ | Ling1S-Top1 | Gemma | HS-QAB (**ours**) | 0.148 ± 0.011 | 0.363 ± 0.032 |
| SciQ | Ling1S-Top1 | Gemma | S-QAB (**ours**) | **0.146 ± 0.006** | 0.366 ± 0.085 |
| SciQ | Ling1S-Top1 | Gemma | S-B | 0.485 ± 0.003 | 0.194 ± 0.014 |
| SciQ | Ling1S-Top1 | Gemma | S | 0.211 ± 0.008 | 0.191 ± 0.008 |
| SciQ | Ling1S-Top1 | Gemma | B | 0.486 ± 0.003 | 0.195 ± 0.011 |
| SciQ | Ling1S-Top1 | Gemma | None | 0.446 ± 0.011 | **0.403 ± 0.018** |
| SciQ | Verb1S-Top1 | Mistral | QAB (**ours**) | 0.262 ± 0.009 | 0.428 ± 0.027 |
| SciQ | Verb1S-Top1 | Mistral | HS-QAB (**ours**) | 0.203 ± 0.006 | **0.474 ± 0.048** |
| SciQ | Verb1S-Top1 | Mistral | S-QAB (**ours**) | **0.182 ± 0.006** | **0.474 ± 0.095** |
| SciQ | Verb1S-Top1 | Mistral | S-B | 0.472 ± 0.003 | 0.237 ± 0.006 |
| SciQ | Verb1S-Top1 | Mistral | S | 0.228 ± 0.005 | 0.27 ± 0.02 |
| SciQ | Verb1S-Top1 | Mistral | B | 0.458 ± 0.002 | 0.377 ± 0.012 |
| SciQ | Verb1S-Top1 | Mistral | None | 0.451 ± 0.013 | 0.451 ± 0.015 |
| SciQ | Verb1S-Top1 | Gemma | QAB (**ours**) | 0.268 ± 0.005 | 0.278 ± 0.021 |
| SciQ | Verb1S-Top1 | Gemma | HS-QAB (**ours**) | 0.194 ± 0.006 | 0.386 ± 0.034 |
| SciQ | Verb1S-Top1 | Gemma | S-QAB (**ours**) | **0.182 ± 0.005** | 0.279 ± 0.023 |
| SciQ | Verb1S-Top1 | Gemma | S-B | 0.477 ± 0.002 | 0.149 ± 0.013 |
| SciQ | Verb1S-Top1 | Gemma | S | 0.243 ± 0.007 | 0.246 ± 0.008 |
| SciQ | Verb1S-Top1 | Gemma | B | 0.464 ± 0.003 | 0.344 ± 0.021 |
| SciQ | Verb1S-Top1 | Gemma | None | 0.495 ± 0.01 | **0.421 ± 0.01** |
| TriviaQA | Ling1S-Top1 | Mistral | QAB (**ours**) | 0.26 ± 0.003 | 0.467 ± 0.024 |
| TriviaQA | Ling1S-Top1 | Mistral | HS-QAB (**ours**) | 0.195 ± 0.007 | 0.499 ± 0.014 |
| TriviaQA | Ling1S-Top1 | Mistral | S-QAB (**ours**) | **0.184 ± 0.005** | **0.541 ± 0.021** |
| TriviaQA | Ling1S-Top1 | Mistral | S-B | 0.441 ± 0.003 | 0.275 ± 0.006 |
| TriviaQA | Ling1S-Top1 | Mistral | S | 0.266 ± 0.005 | 0.28 ± 0.006 |
| TriviaQA | Ling1S-Top1 | Mistral | B | 0.432 ± 0.003 | 0.441 ± 0.009 |
| TriviaQA | Ling1S-Top1 | Mistral | None | 0.326 ± 0.008 | 0.537 ± 0.01 |
| TriviaQA | Ling1S-Top1 | Gemma | QAB (**ours**) | 0.268 ± 0.007 | 0.310 ± 0.034 |
| TriviaQA | Ling1S-Top1 | Gemma | HS-QAB (**ours**) | 0.188 ± 0.02 | **0.359 ± 0.035** |
| TriviaQA | Ling1S-Top1 | Gemma | S-QAB (**ours**) | **0.18 ± 0.005** | 0.351 ± 0.051 |
| TriviaQA | Ling1S-Top1 | Gemma | S-B | 0.235 ± 0.015 | 0.310 ± 0.007 |
| TriviaQA | Ling1S-Top1 | Gemma | S | 0.211 ± 0.012 | 0.086 ± 0.005 |
| TriviaQA | Ling1S-Top1 | Gemma | B | 0.468 ± 0.003 | 0.199 ± 0.019 |
| TriviaQA | Ling1S-Top1 | Gemma | None | 0.486 ± 0.01 | 0.349 ± 0.007 |
| TriviaQA | Verb1S-Top1 | Mistral | QAB (**ours**) | 0.242 ± 0.006 | **0.618 ± 0.016** |
| TriviaQA | Verb1S-Top1 | Mistral | HS-QAB (**ours**) | 0.191 ± 0.01 | 0.573 ± 0.024 |
| TriviaQA | Verb1S-Top1 | Mistral | S-QAB (**ours**) | **0.18 ± 0.006** | 0.545 ± 0.029 |
| TriviaQA | Verb1S-Top1 | Mistral | S-B | 0.436 ± 0.005 | 0.324 ± 0.007 |
| TriviaQA | Verb1S-Top1 | Mistral | S | 0.27 ± 0.005 | 0.352 ± 0.014 |
| TriviaQA | Verb1S-Top1 | Mistral | B | 0.409 ± 0.005 | 0.525 ± 0.019 |
| TriviaQA | Verb1S-Top1 | Mistral | None | 0.38 ± 0.01 | 0.582 ± 0.011 |
| TriviaQA | Verb1S-Top1 | Gemma | QAB (**ours**) | 0.249 ± 0.006 | 0.381 ± 0.02 |
| TriviaQA | Verb1S-Top1 | Gemma | HS-QAB (**ours**) | 0.192 ± 0.006 | **0.417 ± 0.034** |
| TriviaQA | Verb1S-Top1 | Gemma | S-QAB (**ours**) | **0.183 ± 0.008** | 0.337 ± 0.038 |
| TriviaQA | Verb1S-Top1 | Gemma | S-B | 0.444 ± 0.003 | 0.231 ± 0.012 |
| TriviaQA | Verb1S-Top1 | Gemma | S | 0.258 ± 0.008 | 0.225 ± 0.006 |
| TriviaQA | Verb1S-Top1 | Gemma | B | 0.429 ± 0.004 | 0.37 ± 0.015 |
| TriviaQA | Verb1S-Top1 | Gemma | None | 0.527 ± 0.008 | **0.417 ± 0.008** |

Table 6: (Continuation of Table 2) Performance of our QA-calibration methods, QA binning (QAB), Pooled scaling QA binning (S-QAB) and Hierarchical scaling QA binning (HS-QAB)), compared to the baselines, UMD, (B, (Gupta & Ramdas, 2021)), Platt Scaling (S, (Platt, 1999)), and Scaling-binning (S-B, (Kumar et al., 2019)).

| Dataset | Prompt | LLM | Calibrator | CE$(h; \beta)$ | AUAC |
|---------|--------|-----|------------|----------------|------|
| OpenBookQA | Ling1S-Top1 | Mistral | QAB (**ours**) | $0.285 \pm 0.009$ | $0.353 \pm 0.043$ |
| OpenBookQA | Ling1S-Top1 | Mistral | HS-QAB (**ours**) | $0.211 \pm 0.015$ | $0.37 \pm 0.065$ |
| OpenBookQA | Ling1S-Top1 | Mistral | S-QAB (**ours**) | $\mathbf{0.193 \pm 0.012}$ | $0.34 \pm 0.062$ |
| OpenBookQA | Ling1S-Top1 | Mistral | S-B | $0.477 \pm 0.002$ | $0.135 \pm 0.004$ |
| OpenBookQA | Ling1S-Top1 | Mistral | S | $0.32 \pm 0.006$ | $0.133 \pm 0.005$ |
| OpenBookQA | Ling1S-Top1 | Mistral | B | $0.471 \pm 0.004$ | $0.255 \pm 0.023$ |
| OpenBookQA | Ling1S-Top1 | Mistral | None | $0.441 \pm 0.013$ | $\mathbf{0.38 \pm 0.015}$ |
| OpenBookQA | Ling1S-Top1 | Gemma | QAB (**ours**) | $0.316 \pm 0.01$ | $0.25 \pm 0.028$ |
| OpenBookQA | Ling1S-Top1 | Gemma | HS-QAB (**ours**) | $0.221 \pm 0.016$ | $0.261 \pm 0.032$ |
| OpenBookQA | Ling1S-Top1 | Gemma | S-QAB (**ours**) | $\mathbf{0.215 \pm 0.011}$ | $0.257 \pm 0.082$ |
| OpenBookQA | Ling1S-Top1 | Gemma | S-B | $0.415 \pm 0.005$ | $0.129 \pm 0.009$ |
| OpenBookQA | Ling1S-Top1 | Gemma | S | $0.421 \pm 0.01$ | $0.152 \pm 0.013$ |
| OpenBookQA | Ling1S-Top1 | Gemma | B | $0.483 \pm 0.003$ | $0.12 \pm 0.017$ |
| OpenBookQA | Ling1S-Top1 | Gemma | None | $0.434 \pm 0.015$ | $\mathbf{0.31 \pm 0.017}$ |
| OpenBookQA | Verb1S-Top1 | Mistral | QAB (**ours**) | $0.288 \pm 0.008$ | $0.338 \pm 0.035$ |
| OpenBookQA | Verb1S-Top1 | Mistral | HS-QAB (**ours**) | $0.232 \pm 0.012$ | $\mathbf{0.344 \pm 0.018}$ |
| OpenBookQA | Verb1S-Top1 | Mistral | S-QAB (**ours**) | $\mathbf{0.225 \pm 0.011}$ | $0.316 \pm 0.027$ |
| OpenBookQA | Verb1S-Top1 | Mistral | S-B | $0.445 \pm 0.006$ | $0.331 \pm 0.011$ |
| OpenBookQA | Verb1S-Top1 | Mistral | S | $0.282 \pm 0.01$ | $0.3 \pm 0.009$ |
| OpenBookQA | Verb1S-Top1 | Mistral | B | $0.447 \pm 0.003$ | $0.28 \pm 0.02$ |
| OpenBookQA | Verb1S-Top1 | Mistral | None | $0.561 \pm 0.018$ | $0.311 \pm 0.021$ |
| OpenBookQA | Verb1S-Top1 | Gemma | S-QAB (**ours**) | $\mathbf{0.228 \pm 0.011}$ | $\mathbf{0.343 \pm 0.027}$ |
| OpenBookQA | Verb1S-Top1 | Gemma | S-B | $0.410 \pm 0.006$ | $0.301 \pm 0.011$ |
| OpenBookQA | Verb1S-Top1 | Gemma | S | $0.298 \pm 0.01$ | $0.280 \pm 0.011$ |
| OpenBookQA | Verb1S-Top1 | Gemma | B | $0.447 \pm 0.003$ | $0.293 \pm 0.04$ |
| OpenBookQA | Verb1S-Top1 | Gemma | None | $0.561 \pm 0.018$ | $0.315 \pm 0.036$ |
| OpenBookQA | Verb1S-Top1 | Gemma | QAB (**ours**) | $0.261 \pm 0.008$ | $0.332 \pm 0.048$ |
| . OpenBookQA | Verb1S-Top1 | Gemma | HS-QAB (**ours**) | $\mathbf{0.228 \pm 0.012}$ | $0.321 \pm 0.011$ |
| BigBench | Ling1S-Top1 | Mistral | QAB (**ours**) | $0.195 \pm 0.002$ | $0.622 \pm 0.009$ |
| BigBench | Ling1S-Top1 | Mistral | HS-QAB (**ours**) | $\mathbf{0.138 \pm 0.006}$ | $\mathbf{0.690 \pm 0.031}$ |
| BigBench | Ling1S-Top1 | Mistral | S-QAB (**ours**) | $0.157 \pm 0.010$ | $0.641 \pm 0.02$ |
| BigBench | Ling1S-Top1 | Mistral | S-B | $0.387 \pm 0.006$ | $0.523 \pm 0.006$ |
| BigBench | Ling1S-Top1 | Mistral | S | $0.224 \pm 0.003$ | $0.52 \pm 0.004$ |
| BigBench | Ling1S-Top1 | Mistral | B | $0.379 \pm 0.003$ | $0.612 \pm 0.017$ |
| BigBench | Ling1S-Top1 | Mistral | None | $0.245 \pm 0.004$ | $0.684 \pm 0.007$ |
| BigBench | Ling1S-Top1 | Gemma | QAB (**ours**) | $0.198 \pm 0.006$ | $0.41 \pm 0.012$ |
| BigBench | Ling1S-Top1 | Gemma | HS-QAB (**ours**) | $\mathbf{0.142 \pm 0.009}$ | $\mathbf{0.421 \pm 0.021}$ |
| BigBench | Ling1S-Top1 | Gemma | S-QAB (**ours**) | $0.193 \pm 0.002$ | $0.321 \pm 0.052$ |
| BigBench | Ling1S-Top1 | Gemma | S-B | $0.326 \pm 0.002$ | $0.302 \pm 0.005$ |
| BigBench | Ling1S-Top1 | Gemma | S | $0.180 \pm 0.006$ | $0.298 \pm 0.005$ |
| BigBench | Ling1S-Top1 | Gemma | B | $0.468 \pm 0.001$ | $0.293 \pm 0.014$ |
| BigBench | Ling1S-Top1 | Gemma | None | $0.427 \pm 0.006$ | $0.412 \pm 0.01$ |
| BigBench | Verb1S-Top1 | Mistral | QAB (**ours**) | $0.161 \pm 0.003$ | $0.665 \pm 0.031$ |
| BigBench | Verb1S-Top1 | Mistral | HS-QAB (**ours**) | $\mathbf{0.132 \pm 0.009}$ | $0.672 \pm 0.079$ |
| BigBench | Verb1S-Top1 | Mistral | S-QAB (**ours**) | $0.134 \pm 0.004$ | $\mathbf{0.679 \pm 0.012}$ |
| BigBench | Verb1S-Top1 | Mistral | S-B | $0.361 \pm 0.006$ | $0.513 \pm 0.016$ |
| BigBench | Verb1S-Top1 | Mistral | S | $0.208 \pm 0.004$ | $0.522 \pm 0.018$ |
| BigBench | Verb1S-Top1 | Mistral | B | $0.351 \pm 0.002$ | $0.592 \pm 0.014$ |
| BigBench | Verb1S-Top1 | Mistral | None | $0.248 \pm 0.005$ | $0.671 \pm 0.005$ |
| BigBench | Verb1S-Top1 | Gemma | QAB (**ours**) | $0.171 \pm 0.003$ | $\mathbf{0.503 \pm 0.019}$ |
| BigBench | Verb1S-Top1 | Gemma | HS-QAB (**ours**) | $\mathbf{0.159 \pm 0.006}$ | $0.502 \pm 0.011$ |
| BigBench | Verb1S-Top1 | Gemma | S-QAB (**ours**) | $0.215 \pm 0.005$ | $0.465 \pm 0.024$ |
| BigBench | Verb1S-Top1 | Gemma | S-B | $0.456 \pm 0.001$ | $0.289 \pm 0.011$ |
| BigBench | Verb1S-Top1 | Gemma | S | $0.259 \pm 0.004$ | $0.292 \pm 0.007$ |
| BigBench | Verb1S-Top1 | Gemma | B | $0.431 \pm 0.002$ | $0.441 \pm 0.014$ |
| BigBench | Verb1S-Top1 | Gemma | None | $0.458 \pm 0.01$ | $0.499 \pm 0.01$ |

Table 7: (Continuation of Table 2) Performance of our QA-calibration methods, QA binning (QAB), Pooled scaling QA binning (S-QAB) and Hierarchical scaling QA binning (HS-QAB)), compared to the baselines, UMD, (B, (Gupta & Ramdas, 2021)), Platt Scaling (S, (Platt, 1999)), and Scaling-binning (S-B, (Kumar et al., 2019)).

## C    KD-TREE CONSTRUCTION FOR QA-CALIBRATION

We adapt kd-tree (Bentley, 1975) to construct our $\beta$ partitions. We first recall the standard construction of a kd-tree. Let $Z_i = (Z_{i1}, \ldots, Z_{im}, \ldots, Z_{iM}), i = 1, \ldots, N$ be the $M$-dimensional dataset to bin and let $H^k$ with $k = 0$ represent $\{Z_1, \ldots, Z_N\}$. Let $z_m^k$ denote the median of all the $m$th coordinates of the $Z$s in $H^k$. Let $Z[p]$ denote the $p$th coordinate of a $Z$ vector. Set base case value of $k = 0$.

The partitioning scheme to be applied recursively is as follows: split $H^k$ into two halfspaces by pivoting on $z_m^k$, to obtain $H^{2k+1} := \{Z \in H^k : Z[p] \leq z_m^k\}$ and $H^{2k+2} := \{Z \in H^k : Z[p] > z_m^k\}$. The coordinate index $m$ is set to $\lfloor \log 2 \rfloor (k + 1) + 1$, i.e., as we split the nodes, we only change the coordinate index when we switch to another level. When we reach $m = M$, we reset $m = 0$.

In our setting, we form bounded spaces so that our calibrators that contain a QA binning subroutine can generalize well during test time (outliers that are the closest to the boundary of a space will not be assigned to that space). We take the observations with the smallest and largest order statistic in each coordinate used for pivoting, and use them as bounding values for that coordinate.

At test-time, an $M$-vector instance is inserted into one of the leaves by following the same rule as in the partitioning scheme. If any coordinate value is outside its bounding values, assign it to none of the leaves. Otherwise, cycle through the $M$ coordinates and assign the instance to the left tree if the coordinate is less than or equal to $z_m^k$, or to the right tree otherwise. Repeat until a leaf is reached.

## D    GENERALIZING MAXIMUM CALIBRATION ERROR (MCE) TO $\beta$ MAXIMUM CALIBRATION ERROR (QA MCE)

Guo et al. (2017) defined the maximum calibration error (MCE) as:
$$\text{MCE}(h; \beta) = \sup_{r \in \text{range}(h)} [|\mathbb{E}[Y \mid h(Q, A) = r] - r|].$$

We generalize this definition by incorporating $\beta$:

**Definition D.1** (QA Maximum Calibration Error).  The QA maximum calibration error (QA MCE) of $h$ is defined as:
$$\beta\text{-MCE}(h; \beta) = \max_{s \in \text{range}(\beta)} \sup_{r \in \text{range}(h)} [|\mathbb{E}[Y \mid h(Q, A) = r, \beta(Q, A) = s] - r|].$$

This metric assesses the maximum deviation between the confidences and true probabilities over all confidences and partitions. Intuitively, the confidence $r$ that achieves the supremum in MCE may differ from the one that maximizes over partitions in $\beta$-MCE, as the latter is also conditioned on the specific partition.

## E    QUALITATIVE EXAMPLES OF PARTITION CONTENT

Examples of question-and-answer pairs assigned to $\beta$ partitions are presented in Table 8.

## F    EXTRA EXPERIMENTS WITH VARYING $\beta$ CHOICES

In addition to using kd-tree as $\beta_{\text{bin}}$ and DistilBERT embedding as $\beta_{\text{emb}}$, we also use k-means as $\beta_{\text{bin}}$ and XLNet (Yang et al., 2019) embeddings as $\beta_{\text{emb}}$ We utilize the [CLS] token embedding from a pre-trained XLNet and conduct a hyperparameter search over the number of centroids, using AUAC as the validation metric. We present the results in Table 9. Similar patterns to those in Table 2 are observed: our methods, S-QAB and HS-QAB, achieve the highest performance for $\text{CE}(h; \beta)$ and AUAC.

## G    SCATTERPLOT OF RECALIBRATED CONFIDENCE SCORES OVER THE CONFIDENCE ELICITATION BASELINE

We examine how the recalibrated confidence scores differ for baseline confidence scores obtained using the Ling1STop1 (Figure 3). The confidence scores obtained by our method QA binning have

| Question | LLM Answer |
|---|---|
| If weather is stormy then there is a greater chance of | everything getting wet |
| Bill missed high tide, so he had to wait until when to see it again | tomorrow |
| Great lakes may have come to be thanks to | ice pillars |
| If a flood is occurring, there was most likely | great droplets repeating |
| Some animals are aided in finding food sources by | aroma |
| Flowers make themselves attractive to hummingbirds with | an optimal angle |
| Hummingbirds gather nectar using their | bills |
| An example of an adult animal laying eggs is all aside from | kittens |
| When a human's organs stop working and he stops breathing, that person | perished |
| If a thing experiences a burning combustion, then it is | damaged |
| An exertion on a thing that is going against the thing's intended direction, when in motion will | oppose it |
| When we think of bees, we also think of pollen. This is because bees | consume it |

Table 8: Qualitative examples of three random partitions and four random question-and-answer pairs from the OpenBookQA dataset, a multiple-choice dataset. The prompt used is Ling1STop1 (Table 5) and answers are generated using Mistral. The partitioning function $\beta$ leverages DistilBERT embeddings and a kd-tree with a maximum depth of 7. The underlying topic for the first two partitions seem clear: "water" and "animal", and the third partition is likely to have identified the form of the question.

| Dataset | Prompt | LLM | Calibrator | $\beta_{\text{emb}}$ | $\beta_{\text{bin}}$ | CE$(h; \beta)$ | AUAC |
|---|---|---|---|---|---|---|---|
| OpenBookQA | Verb1S-Top1 | Mistral | QAB (**ours**) | DistilBERT | kd-tree | $0.288 \pm 0.008$ | $0.338 \pm 0.035$ |
| OpenBookQA | Verb1S-Top1 | Mistral | HS-QAB (**ours**) | DistilBERT | kd-tree | $0.232 \pm 0.012$ | $\mathbf{0.344 \pm 0.018}$ |
| OpenBookQA | Verb1S-Top1 | Mistral | S-QAB (**ours**) | DistilBERT | kd-tree | $\mathbf{0.225 \pm 0.011}$ | $0.316 \pm 0.027$ |
| OpenBookQA | Verb1S-Top1 | Mistral | S-B | DistilBERT | kd-tree | $0.445 \pm 0.006$ | $0.331 \pm 0.011$ |
| OpenBookQA | Verb1S-Top1 | Mistral | S | DistilBERT | kd-tree | $0.282 \pm 0.01$ | $0.3 \pm 0.009$ |
| OpenBookQA | Verb1S-Top1 | Mistral | B | DistilBERT | kd-tree | $0.447 \pm 0.003$ | $0.28 \pm 0.02$ |
| OpenBookQA | Verb1S-Top1 | Mistral | None | DistilBERT | kd-tree | $0.561 \pm 0.018$ | $0.311 \pm 0.021$ |
| OpenBookQA | Verb1S-Top1 | Mistral | QAB (**ours**) | DistilBERT | k-means | $0.310 \pm 0.009$ | $0.334 \pm 0.012$ |
| OpenBookQA | Verb1S-Top1 | Mistral | HS-QAB (**ours**) | DistilBERT | k-means | $\mathbf{0.240 \pm 0.013}$ | $\mathbf{0.340 \pm 0.019}$ |
| OpenBookQA | Verb1S-Top1 | Mistral | S-QAB (**ours**) | DistilBERT | k-means | $0.255 \pm 0.012$ | $0.321 \pm 0.019$ |
| OpenBookQA | Verb1S-Top1 | Mistral | S-B | DistilBERT | k-means | $0.450 \pm 0.007$ | $0.331 \pm 0.011$ |
| OpenBookQA | Verb1S-Top1 | Mistral | S | DistilBERT | k-means | $0.285 \pm 0.011$ | $0.3 \pm 0.009$ |
| OpenBookQA | Verb1S-Top1 | Mistral | B | DistilBERT | k-means | $0.452 \pm 0.004$ | $0.28 \pm 0.02$ |
| OpenBookQA | Verb1S-Top1 | Mistral | None | DistilBERT | k-means | $0.565 \pm 0.018$ | $0.311 \pm 0.021$ |
| OpenBookQA | Verb1S-Top1 | Mistral | QAB (**ours**) | XLNet | kd-tree | $0.295 \pm 0.010$ | $0.337 \pm 0.034$ |
| OpenBookQA | Verb1S-Top1 | Mistral | HS-QAB (**ours**) | XLNet | kd-tree | $0.238 \pm 0.012$ | $0.341 \pm 0.021$ |
| OpenBookQA | Verb1S-Top1 | Mistral | S-QAB (**ours**) | XLNet | kd-tree | $\mathbf{0.220 \pm 0.011}$ | $\mathbf{0.347 \pm 0.017}$ |
| OpenBookQA | Verb1S-Top1 | Mistral | S-B | XLNet | kd-tree | $0.442 \pm 0.008$ | $0.331 \pm 0.011$ |
| OpenBookQA | Verb1S-Top1 | Mistral | S | XLNet | kd-tree | $0.280 \pm 0.012$ | $0.3 \pm 0.009$ |
| OpenBookQA | Verb1S-Top1 | Mistral | B | XLNet | kd-tree | $0.450 \pm 0.004$ | $0.28 \pm 0.02$ |
| OpenBookQA | Verb1S-Top1 | Mistral | None | XLNet | kd-tree | $0.561 \pm 0.018$ | $0.311 \pm 0.021$ |
| OpenBookQA | Verb1S-Top1 | Mistral | QAB (**ours**) | XLNet | k-means | $0.312 \pm 0.011$ | $0.335 \pm 0.033$ |
| OpenBookQA | Verb1S-Top1 | Mistral | HS-QAB (**ours**) | XLNet | k-means | $0.246 \pm 0.013$ | $0.342 \pm 0.019$ |
| OpenBookQA | Verb1S-Top1 | Mistral | S-QAB (**ours**) | XLNet | k-means | $\mathbf{0.235 \pm 0.010}$ | $\mathbf{0.350 \pm 0.018}$ |
| OpenBookQA | Verb1S-Top1 | Mistral | S-B | XLNet | k-means | $0.448 \pm 0.007$ | $0.331 \pm 0.011$ |
| OpenBookQA | Verb1S-Top1 | Mistral | S | XLNet | k-means | $0.283 \pm 0.011$ | $0.3 \pm 0.009$ |
| OpenBookQA | Verb1S-Top1 | Mistral | B | XLNet | k-means | $0.455 \pm 0.005$ | $0.28 \pm 0.02$ |
| OpenBookQA | Verb1S-Top1 | Mistral | None | XLNet | k-means | $0.565 \pm 0.018$ | $0.311 \pm 0.021$ |

Table 9: We vary $\beta_{\text{emb}} \in \{\text{DistilBERT}, \text{XLNet}\}$ and $\beta_{\text{bin}} \in \{\text{kd-tree}, \text{k-means}\}$. Note that AUAC metrics for S-B, S, B and None remain constant as they do not utilize $\beta$. In this experiment, we observe that across the four variations listed above, the optimal $\beta_{\text{bin}}$ results in a comparable number of partitions.

higher sharpness (Brier, 1950; Gneiting et al., 2007) since we are fitting a UMD fitter per partition. The baseline UMD (B), shown in blue, maps an elicited confidence score from Ling1STop1 to either a single value or a small set of nearby values, due to the delta-randomization in UMD. In contrast, our method, QA binning (QAB), shown in grey, applies a partition-specific UMD that adapts to the specific question-and-answer pair. As a result, the scores calibrated by QA binning span a broader range. This is intuitively reasonable, as a prompt-elicited confidence score of 0.1 from a topic like "geography" may require a different recalibration compared to the one from the "politics".

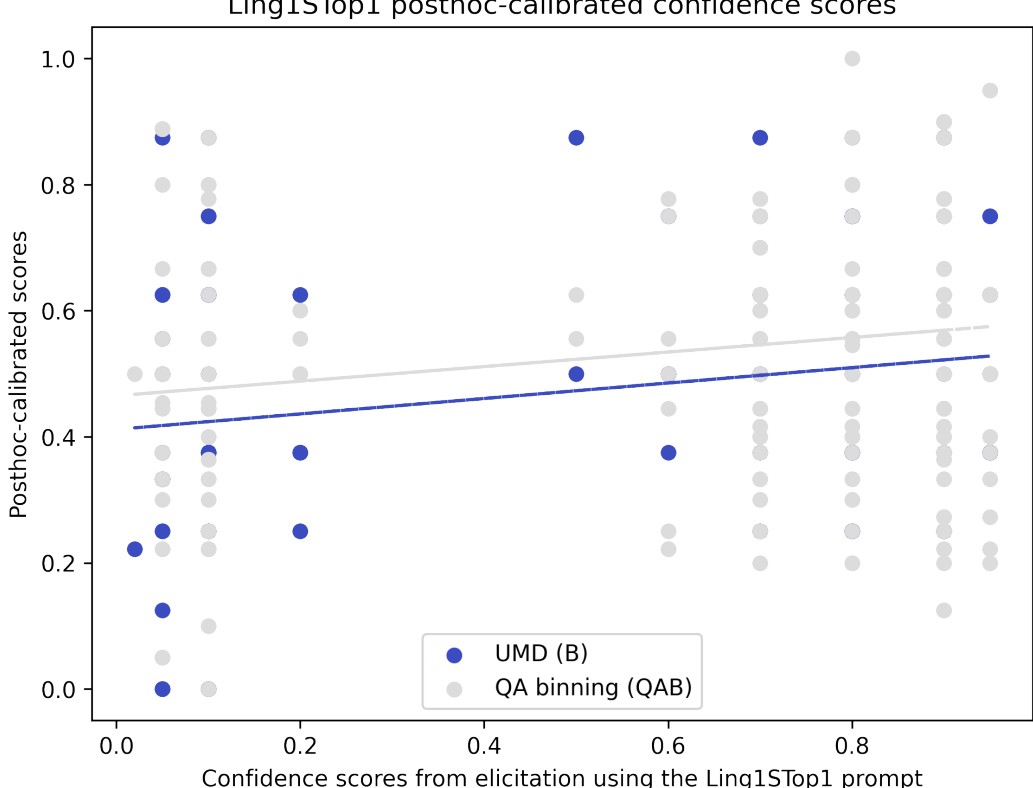

Figure 3: Scatter plot and regression lines for posthoc-calibrated scores vs confidence scores from Ling1STop1. Note that confidence scores from Ling1STop1 (shown in the the x-axis) are discretized since confidence statement is drawn from an expression list (Table 5). We use the OpenBookQA dataset and Mistral LLM.

## H RELIABILITY ANALYSIS OF QA PARTITIONED DATA

In this section, we demonstrate reliability analysis using ECE and reliability plot through an example. We perform an analysis for each of the four $\beta$ partitions (derived using a kd-tree with a maximum depth of two) in Figure 4. We compare the baselines None (from Ling1S-Top1 prompt) and UMD (which is known to best optimize ECE) against our method QA binnning. Partition-wise analysis reveals that the ECE for each partition (representing question-and-answer pairs that are semantically similar) is best achieved by our method, QA binning. However, performing this reliability analysis becomes increasingly challenging and unwieldy as the cardinality of the range of $\beta$ grows.

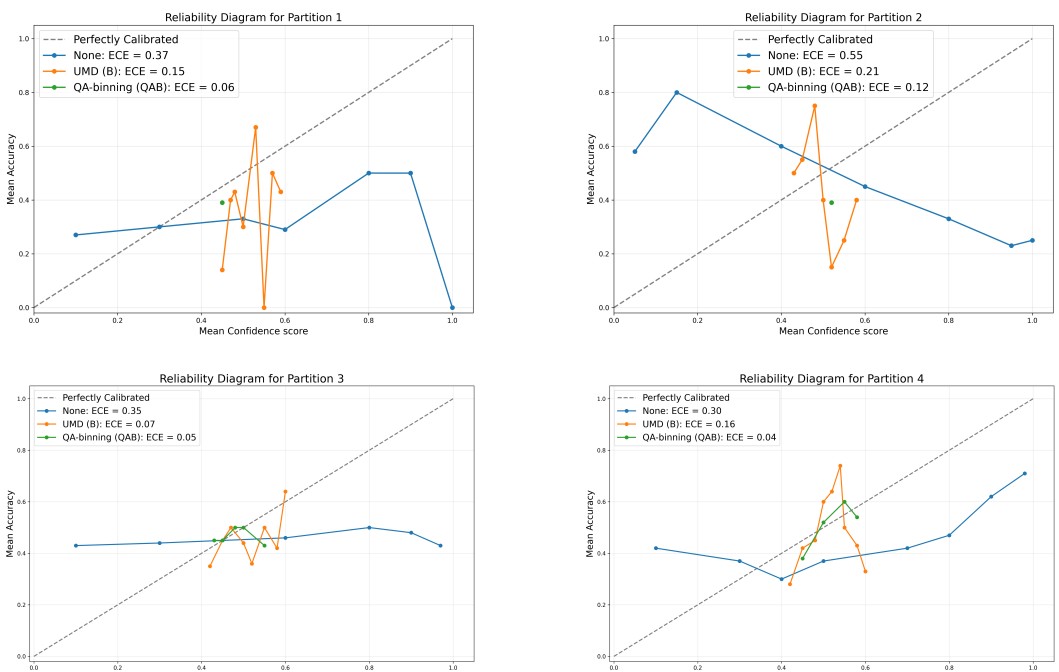

Figure 4: The reliability plot compares the baselines None , UMD (B), and our method, QA binning (QAB), across each QA partition. Using the OpenBookQA dataset, Ling1S-Top1 prompt, a Mistral LLM, and a kd-tree with a maximum depth of two, four QA partitions are generated, and we conduct an analysis on each partition. UMD (B) and QA binning (QAB), provide confidence scores that are better calibrated at each of the partition compared to None. However, QA binning (QAB) yields better calibrated confidence scores than UMD (B).

