# OpenReview forum: "QA-Calibration of Language Model Confidence Scores"
_ICLR.cc/2025/Conference — ICLR 2025 Poster_

### Official Review · Reviewer_iAXj · 2024-11-03

**Soundness:** 4
**Presentation:** 3
**Contribution:** 3
**Rating:** 8
**Confidence:** 3

**Summary:**

The paper introduces a stronger form of calibration which requires that subgroups also be calibrated. It then presents a set of techniques for achieving this form of calibration, first by binning *within* the subgroups and then introducing two improvements: scaling by replacing the labels with predicted labels during the calibration step (s-beta-binning) and then replacing the predictor with a hierarchical classifier to get pooling across groups.

The main theoretical result is that these methods can achieve bounded calibration error with high probability, using a bound in which the error rate grows as $O(\log b / (2b - 1))$ with $b$ equal to the number of bins.

Empirically, these techniques improve calibration error over reasonable baselines on a suite of five question answering tasks; they sometimes improve selection prediction as well.

**Strengths:**

This is a solid contribution that offers a novel and useful new perspective on post-hoc calibration. The methods are well motivated and extend prior work in sensible but interesting ways, which will likely be of interest to other researchers working on calibration. The exposition is unusually clear and theoretical results are helpful.

**Weaknesses:**

The empirical results may not be strong enough to motivate adoption by practitioners -- while the method outperforms baselines in table 2, over the full suite of tasks the selective prediction performance is often worse than or statistically indistinguishable from elicited confidence scores (tables 6 & 7).

A related point from the practitioner's perspective is that the utility of beta-calibration seems to depend on the meaningfulness of the groups -- if the questions really are partitioned by users as suggested by figure 1, then these stronger guarantees seem crucial; but if the groups are not intuitively meaningful then it's not clear why a practitioner would be motivated to pursue these stronger guarantees.

**Questions:**

While the paper is focused on language model confidence and generative QA, is there anything in the method that is specific to this application? It seems like it could be applied to any prediction task over examples that can be partitioned by a KD tree in which recalibration might be required.

So I would love to know more about (a) which examples end up in which bins in these datasets; (b) the relationship between recalibrated confidence scores with and without beta-binning (e.g. something like a scatterplot of confidence scores with "B" vs "BB").

---

### Official Review · Reviewer_YLZ3 · 2024-11-04

**Soundness:** 3
**Presentation:** 2
**Contribution:** 3
**Rating:** 6
**Confidence:** 5

**Summary:**

This paper deals with post-hoc calibration of “verbalized” confidences from LLMs, which are typically not well-calibrated out-of-the-box. The paper uses as a subroutine, uniform-mass double-dipping histogram binning (UMD) (Gupta & Ramdas, 2021) to perform this post-hoc calibration. A further scaling version is explored in S3.2 based on the scaling procedure from Kumar et al. (2019). The proposed approach is compared to existing (general purpose) post-hoc calibration methods on standard QA benchmarks.

**Strengths:**

* The paper tackles an important problem (LLM confidence estimation)
* The approach is validated on 5 different QA datasets.
* The “selective” QA metric provides some insight into the practical benefit of improved calibration.

**Weaknesses:**

* The approach assumes access to an “oracle” to measure semantic equivalence between a predicted and reference answer. I’m not sure if Llama-3.1 is such an oracle, unless perhaps it has memorized the QA datasets in question, which raises some other concerns.
The approach requires dataset-specific calibration sets. I’m not sure if it’s fair to compare to baselines such as out-of-the-box prompts that don’t use this information. It’s also a bit of a limitation since it presumably makes it difficult (impossible) to apply the proposed approach to novel problems.
* Overall, the approach seems to largely be an application of existing methods to post-hoc calibration of a particular QA problem with LLMs. The adaptation to LLMs does not appear to involve any significant technical innovations.
* The paper evaluates the results using the very calibration metric that is optimised by the proposed method. This is fraught and it’s unclear why further calibration metrics (such as one or more variations of ECE, Brier score, etc.) are not also reported.
* The approach relies on a smaller/older model DistilBERT which seems like a key limitation.
* While there is a short paragraph addressing the limitation of choosing \Beta, I would have liked to see further more in-depth discussion of the limitations of the paper.

**Questions:**

* Why not include additional metrics such as ECE and Brier score?

---

### Official Review · Reviewer_cg5J · 2024-11-08

**Soundness:** 2
**Presentation:** 3
**Contribution:** 2
**Rating:** 5
**Confidence:** 4

**Summary:**

This work presents a beta calibration error, a calibration metric which partitioning the test set and computing calibration error over each partition individually. Partitioning is done based on sentence representations from a pretrained LM. The authors then present methods for post-hoc calibrating existing confidence estimates from an LM. Their approach involves partitioning training data using the same partitioning function as is used during evaluation and fitting calibrators for each partition.

**Strengths:**

This work defines a generalization of calibration error metrics.

The work introduces methods for optimizing for their proposed calibration metric and theoretical results backing up their methods.

**Weaknesses:**

W1. One concern with its work is a possible mischaracterization of how expected calibration error (ECE) is used to evaluate calibration. In the toy example in Table 1, the authors note that standard calibration error is 0; however, the often used ECE metric (from [1]) involves binning examples during test time by sorting and partitioning examples based on their predicted confidence. In the Toy example in Figure 1, ECE  (with 2 bins) and beta-calibration are equivalent. While Beta-Calibration can be framed as a generalization of the standard ECE binning method, it's unclear why it's better motivated or more useful.

[1] Obtaining Well Calibrated Probabilities Using Bayesian Binning
Mahdi Pakdaman Naeini, Gregory F. Cooper, Milos Hauskrecht

W2. Following up on point 1, while it can make sense to introduce alternative binning when computing ECE, by using different beta-calibration functions, it is not made clear why the proposed binning method is inherently better and why methods should be optimized for the specific binning function used during evaluation.

W3. Another weakness is in using only one base calibration method. Furthermore, while the authors use verbalized confidence scores as their base confidence scores, citing them as a SOTA method, it's unclear whether this is true for the scale of model evaluated in this work (side note, work also lacks core modeling details like the size of pretrained LMs evaluated and whether they use instruction-tuned variants).

**Questions:**

As an additional note MCE (noted in related work) was also introduced by [1]. Frequently used metrics like AUROC and other metrics should be also included in related work and discussed, many are summarized by [2].

[2] Plex: Towards Reliability using Pretrained Large Model Extensions
Dustin Tran, Jeremiah Liu, Michael W. Dusenberry, Du Phan, Mark Collier, Jie Ren, Kehang Han, Zi Wang, Zelda Mariet, Huiyi Hu, Neil Band, Tim G. J. Rudner, Karan Singhal, Zachary Nado, Joost van Amersfoort, Andreas Kirsch, Rodolphe Jenatton, Nithum Thain, Honglin Yuan, Kelly Buchanan, Kevin Murphy, D. Sculley, Yarin Gal, Zoubin Ghahramani, Jasper Snoek, Balaji Lakshminarayanan

---

### Official Review · Reviewer_snQU · 2024-11-09

**Soundness:** 3
**Presentation:** 3
**Contribution:** 3
**Rating:** 8
**Confidence:** 3

**Summary:**

The paper introduces a novel concept called $\beta$-calibration, which calculates calibration errors across various QA groups to enhance user decision-making. It then proposes post-hoc calibration techniques specifically tailored for $\beta$-calibration, establishing guarantees without additional assumptions. Experimental results highlight the effectiveness of $\beta$-calibration across a range of tasks and models, demonstrating improvements in calibration performance.

**Strengths:**

1. The problem of calibration dependency on the dataset is highly significant. The QA pairs in different subsets can exhibit vastly different subset calibration errors, affecting the user's experience. To the best of my knowledge, this is the first paper to identify and directly address this issue. While the proposed methods are relatively straightforward adaptations of existing approaches, they represent an essential first step in tackling this problem. Although initial, this approach is an important contribution.
2. The paper is well-structured and easy to follow. Notations are defined with clarity and rigor, enhancing overall comprehension.
3. The experiments are thoughtfully designed, evaluating not only calibration error but also the utility of $\beta$-calibration in selective answering, demonstrating that the model is both calibrated and practically useful. Experiments are conducted across diverse models and tasks, with results showing significant improvements over baseline methods. Additionally, the inclusion of rigorously proven distribution-free guarantees completes the study.

**Weaknesses:**

1. **Generalizability of $\beta$:** The paper is motivated by the idea that users may have specific interests in different groups of QAs, necessitating calibration that is tailored to user needs. However,
- The paper demonstrates the method’s effectiveness using only one type of $\beta$ (partitioning based on DistillBERT embeddings), leaving it unclear how well these methods generalize to other partitioning strategies. Showing results with other partitioning methods or justifying this choice’s relevance to user interests would strengthen the paper.
- The framework assumes each question belongs exclusively to one group. However, in real-world scenarios, a single question could naturally align with multiple groups, as users with differing interests may ask the same question. It is unclear how the framework would function under this condition. Discussing this limitation or proposing a solution for handling questions that span multiple user-interest groups would enhance the paper’s practical value.
2. **Method Choice Motivation:** The framework, which partitions then calibrates each subset, could work with various post-hoc methods (e.g., isotonic regression, Platt scaling), yet only binning is tested. Including experiments to test the framework with different post-hoc strategies or discuss the reasons for this choice could strengthen the paper by demonstrating its generalizability. Additionally, it is unclear why isotonic regression—a common baseline—is excluded.
3. **Limited Result Analysis:** While partition-specific calibration errors are a focus, results are only reported on average. Reporting variance across partitions, or providing results for each partition in an additional table, along with identifying and discussing those showing the most improvement, would offer valuable insights.

**Questions:**

1. In Equation (5), how are the fixed and random intercepts/slopes calculated separately? It appears possible to compute an intercept-slope pair for each partition, so how do you differentiate between fixed and random components? Additionally, how does this approach differ from simply calculating an intercept-slope pair for each partition?
2. What value of $\alpha$ was used when plotting Figure 2 and how is that chosen?
3. When calculating calibration error, how are the bins constructed? Do you use the same binning strategy as the binning method in the paper, or is a fixed-width re-binning applied?
4. How tight is the bound compared with the actual results? How does the bound informs the selection of hyper-parameter? It seems like the choice of B follows prior work rather than leveraging the newly calculated bound.

Overall, I think this is a good paper. I’m open to raising the score if my concerns are addressed.

---

> ### Comment · Reviewer_snQU · 2024-11-28
> **Thanks for the response**
>
> Thank you for the detailed response, which has addressed most of my concerns. I appreciate the effort put into clarifying the points I raised.  However, I would like to offer two additional suggestions to further enhance the paper. While I do not expect the authors to address these during the rebuttal period, I hope they can be incorporated into the final version if the paper is accepted.
>
> 1. **Alternative Partitioning Strategies**: While the current focus on semantic similarity is valid, I believe exploring alternative partitioning strategies instead of different embedding models, such as domain-based, event-based, topic-based, or difficulty-based partitions, could provide valuable insights. These approaches, based on entirely different logics from semantic similarity, may better align with real-world applications. I recommend including a discussion on the potential relevance of these partitions, how they might integrate with the proposed framework (or why they might not), and whether they could influence the framework’s effectiveness. Ideally, one experiment using a partitioning logic entirely distinct from semantic similarity would further strengthen the paper.
> 2. **Partition-Based Metric Analysis**: While I understand that the number of partitions might be large, it would be helpful to present a partition-based metric, such as the percentage of partitions that show significant improvement, no improvement, or degradation. This would provide readers with a clearer understanding of whether the proposed method improves most partitions or primarily benefits a few, potentially at the expense of others. Such an analysis would offer a more nuanced view of the method’s overall impact, ensuring it aligns with practical expectations.
>
> The points I raised are more of a bonus for consideration. Given that the current version already feels complete and addresses my primary concerns, I have raised my score accordingly. Happy Thanksgiving!

---

### Meta-Review · Area_Chair_rpPG · 2024-12-23

**Metareview:**

This paper introduces the concept of beta-calibration, a novel framework for calibrating confidence scores across user-defined subgroups in generative QA tasks. The paper proposes post-hoc calibration methods, scaling-beta-binning (S-BB) and hierarchical scaling beta-binning (HS-BB), which are theoretically supported with distribution-free guarantees. The empirical evaluation demonstrates the methods' efficacy across several QA datasets.

*Strengths*:
-Novelty: The concept of β-calibration provides an extension of standard calibration metrics, allowing for subgroup-specific evaluation that aligns with practical applications.
-Theoretical results: The paper provides distribution-free guarantees for the proposed methods/
-Empirical results: Comprehensive experiments demonstrate the methods' effectiveness across various datasets, models, and metrics, including selective QA.
-The selective QA metric provides some insight into the practical benefit of improved calibration.
-Clarity: The paper is well-written and logically structured, with clear mathematical formulations and notations.

*Weaknesses*:
-Limited exploration of partitioning strategies and the approach being limited to smaller/older models (snQU, iAXj, YLZ3)
-Limited analysis
-Evaluation using a metric tied to the proposed framework
-The approach assumes access oracle

Overall, the author response seems to address all the major concerns, and the paper makes strong contributions on post-hoc calibration with a novel perspective.

**Additional Comments On Reviewer Discussion:**

The major concern was about generalizability of results. To address this the author response provides additional experiments with XLNet embeddings and k-means binning to demonstrate the framework's generalizability beyond DistilBERT. The response also includes additional experiments to address remaining concerns. Examples:
-Including results on Isotonic regression
-Experiments simulating weaker oracles showed the framework's robustness even with noisy or imperfect calibration targets.

---

### Decision · Program_Chairs · 2025-01-22

Accept (Poster)